# Scaling Laws for Hyperparameter Optimization

**Arlind Kadra**
Representation Learning Lab
University of Freiburg
kadraa@cs.uni-freiburg.de

**Maciej Janowski**
Representation Learning Lab
University of Freiburg
janowski@cs.uni-freiburg.de

**Martin Wistuba**
Amazon Web Services
Amazon Berlin
marwistu@amazon.com

**Josif Grabocka**
Representation Learning Lab
University of Freiburg
grabocka@cs.uni-freiburg.de

## Abstract

Hyperparameter optimization is an important subfield of machine learning that focuses on tuning the hyperparameters of a chosen algorithm to achieve peak performance. Recently, there has been a stream of methods that tackle the issue of hyperparameter optimization, however, most of the methods do not exploit the dominant power law nature of learning curves for Bayesian optimization. In this work, we propose Deep Power Laws (DPL), an ensemble of neural network models conditioned to yield predictions that follow a power-law scaling pattern. Our method dynamically decides which configurations to pause and train incrementally by making use of gray-box evaluations. We compare our method against 7 state-of-the-art competitors on 3 benchmarks related to tabular, image, and NLP datasets covering 59 diverse tasks. Our method achieves the best results across all benchmarks by obtaining the best any-time results compared to all competitors.

## 1 Introduction

Hyperparameter Optimization (HPO) is a major challenge for the Machine Learning community. Unfortunately, HPO is not yet feasible for Deep Learning (DL) methods due to the high cost of evaluating multiple configurations. Recently multi-fidelity HPO (a sub-problem of gray-box HPO) has emerged as a promising paradigm for HPO in DL, by discarding poorly-performing hyperparameter configurations after observing the validation error on the low-level fidelities of the optimization procedure [28, 9, 1, 29]. The advantage of multi-fidelity HPO compared to online HPO [7, 38], or meta-gradient HPO [32, 10, 31] is the ability to tune all types of hyperparameters.

In recent years, a stream of papers highlights the fact that the performance of DL methods is predictable [14], concretely, that the validation error rate is a power law function of the model size, or dataset size [41, 40]. Such a power law relationship has been subsequently validated in the domain of NLP, too [11]. In this paper, we demonstrate that the power-law principle has the potential to be a game-changer in HPO, because we can evaluate hyperparameter configurations in low-budget regimes (e.g. after a few epochs), then estimate the performance on the full budget using dataset-specific power law models. Concretely, we hypothesize and empirically demonstrate that optimization curves (training epochs versus accuracy, or loss) can be efficiently modeled as simple power law functions.

As a result, we introduce Deep Power Law (DPL) ensembles, a probabilistic surrogate for Bayesian optimization (BO) that estimates the performance of a hyperparameter configuration at future budgets using ensembles of deep power law functions. Subsequently, our novel formulation of BO dynamically decides which configurations to pause and train incrementally by relying on the performance

37th Conference on Neural Information Processing Systems (NeurIPS 2023).

estimations of the surrogate. We demonstrate that our method achieves the new state-of-the-art in HPO for DL by comparing against 7 strong HPO baselines, and 59 datasets of three diverse modalities (tabular, image, and natural language processing). Our experimental protocol contains state-of-the-art deep learning architectures such as Transformers [44], XFormer [26], ResNeXt [49] and large language models like GPT-2 [39]. As a result, we believe the proposed method has the potential to finally make HPO for DL a feasible reality.

Overall, our contributions can be summarized as follows:

- We introduce a novel probabilistic surrogate for multi-fidelity HPO based on ensembles of deep power law functions.
- We derive a simple mechanism to combine our surrogate with Bayesian optimization.
- Finally, we demonstrate the empirical superiority of our method against the current state-of-the-art in HPO for Deep Learning, with a large-scale HPO experimental protocol.

## 2  Related Work

**Multi-fidelity HPO** assumes a method has access to the learning curve of a hyperparameter configuration. Such a learning curve is the function that maps either training time or dataset size, to the validation performance. The early performance of configurations (i.e. first segment of the learning curve) is used to discard unpromising configurations, before waiting for full convergence. Successive halving [17] is a widely used multi-fidelity method that randomly samples hyperparameter configurations, starts evaluating them, and ends a fraction of them upon reaching a predefined budget. Afterwards, the budget is divided by the fraction of discarded hyperparameter configurations and the process continues until the maximum budget is reached. Although the method relies only on the last observed value of the learning curve, it is very efficient. In recent years, various flavors of successive halving have been suggested, including Hyperband [28], which effectively runs successive halving in parallel with different settings. A major improvement to Hyperband is replacing random search with a more efficient sampling strategy [1, 9]. A more efficient approach is to allocate the budget dynamically to the most promising configurations [47]. However, the only assumption these methods make about the learning curve is that it will improve over time, while recent work [4] exploits a power law assumption on the curves. Similarly, we fit surrogates that exploit a power law assumption, however, our method is able to estimate the performance of unobserved configurations through probabilistic surrogates learned jointly for all hyperparameter configurations.

**Learning curve prediction** is a related topic, where the performance of a configuration is predicted based on a partially observed learning curve [36]. Typically, the assumptions about the learning curve are much stronger than those described above. The prediction is often based on the assumption that the performance increases at the beginning and then flattened towards the end. One way to model this behavior is to define a weighted set of parametric functions [8, 23]. Then, the parameters of all functions are determined so that the resulting prediction best matches the observed learning curve. Another approach is to use learning curves from already evaluated configurations and to find an affine transformation that leads to a well-matched learning curve [6]. A more data-driven approach is to learn the typical learning curve behaviour directly from learning curves across different datasets [48]. Learning curve prediction algorithms can be combined with successive halving [2]. In contrast to this research line, we fit ensembles of power law surrogates for conducting multi-fidelity HPO with Bayesian optimization.

**Scaling laws** describe the relationship between the performance of deep learning models as a function of dataset size or model size as a power law [14, 16, 41, 40, 11, 20, 15, 33]. Another work tunes hyperparameters on a small-scale model and then transfers them to a large-scale version [50]. In contrast to these papers, we directly use the power law assumption for training surrogates in Bayesian optimization for HPO.

## 3  Preliminaries

**Hyperparameter Optimization (HPO)** demands finding the configurations $\lambda \in \Lambda$ of a Machine Learning method that achieve the lowest validation loss $\mathcal{L}^{(\text{Val})}$ of a model (e.g. a neural network), which is parameterized with $\theta \in \Theta$ and learned to minimize the training loss $\mathcal{L}^{(\text{Train})}$ as:

$$\lambda^* := \underset{\lambda \in \Lambda}{\arg\min} \ \mathcal{L}^{(Val)}(\lambda, \theta^*(\lambda)), \quad \text{s.t.} \quad \theta^*(\lambda) := \underset{\theta \in \Theta}{\arg\min} \ \mathcal{L}^{(Train)}(\lambda, \theta) \tag{1}$$

For simplicity, we denote the validation loss as our function of interest $f(\lambda) := \mathcal{L}^{(Val)}(\lambda, \theta^*(\lambda))$. The optimal hyperparameter configurations $\lambda^*$ of Equation 1 are found via **an HPO policy** $\mathcal{A} \in \mathcal{P}$ (also called an HPO method) that given a history of $N$ evaluated configurations $H^{(N)} := \{\lambda_i, f(\lambda_i)\}_{i=1}^N$ suggests the $(N+1)$-th configuration to evaluate as $\lambda_{N+1} := \mathcal{A}(H^{(N)})$ where $\mathcal{A} : [\Lambda \times \mathbb{R}_+]^N \to \Lambda$. The search for an optimal HPO policy is a bi-objective problem in itself, aiming at (i) finding a configuration out of $N$ evaluations that achieves the smallest validation loss $f(\lambda)$, and (ii) ensuring that the costs of evaluating the $N$ configurations do not exceed a total budget $\Omega$, as shown in Equation 2.

$$\underset{\mathcal{A} \in \mathcal{P}}{\arg\min} \ \underset{i \in \{1, \dots, N\}}{\min} \ f\left(\lambda_i := \mathcal{A}\left(H^{(i-1)}\right)\right), \tag{2}$$

$$\text{where:} \qquad H^{(i)} := \begin{cases} \{(\lambda_j, f(\lambda_j))\}_{j=1}^i & i > 0 \\ \emptyset & i = 0 \end{cases},$$

$$\text{subject to:} \quad \Omega \geq \sum_{i=1}^N \text{cost}\left(f(\lambda_i)\right)$$

**Bayesian optimization (BO)** is the most popular type of policy for HPO, due to its ability to balance the exploration and exploitation aspects of minimizing the validation loss $f$ in terms of hyperparameters $\lambda \in \Lambda$. Technically speaking, BO fits a surrogate $\hat{f}(\lambda; \phi)$ parametrized with $\phi \in \Phi$ to approximate the observed loss $f(\lambda)$ using the history $H^{(N)}$, as $\phi^* := \arg\max_{\phi \in \Phi} \mathbb{E}_{(\lambda, f(\lambda)) \sim p_{H^{(N)}}} p(f(\lambda)|\lambda, \phi)$, where, $p$ represents the probability. Afterwards, BO uses an acquisition/utility function $a : \Lambda \to \mathbb{R}_+$ to recommend the next configuration as $\lambda_{N+1} := \mathcal{A}(H^{(N)}) = \arg\max_{\lambda \in \Lambda} a\left(\hat{f}(\lambda; \phi^*)\right)$.

**Multi-fidelity HPO** refers to the case where an approximation of the validation loss can be measured at a lower budget $b \in B$, where $B := (0, b_{\max}]$. For instance, in Deep Learning we can measure the validation loss after a few epochs ($0 < b < \epsilon$), rather than wait for a full convergence ($b = b_{\max}$). Throughout this paper, the term budget refers to a learning curve step. The evaluation of a configuration $\lambda$ for a budget $b$ is defined as $f(\lambda, b) : \Lambda \times B \to \mathbb{R}_+$.

## 4 Power Law Surrogates for Bayesian Optimization

Prior work has demonstrated that the performance of Machine Learning methods as a function of budgets (i.e. dataset size, number of optimization epochs, model size, image resolution) follows a power law relationship [41, 40, 11, 20, 15]. In this work, we employ this power law dependence between the validation loss and the number of optimization epochs in Deep Learning. We propose a novel multi-fidelity Hyperparameter Optimization method which is based on power law surrogates. We assume that every learning curve $f(\lambda, \cdot)$ can be described by a power law function defined by $(\alpha, \beta, \gamma)$. Concretely, we define a power law function for the validation loss of a configuration $\lambda$ at a budget $b$ (a.k.a. the number of epochs) as shown in Equation 3.

$$\hat{f}(\lambda, b) := \alpha_\lambda + \beta_\lambda \ b^{-\gamma_\lambda}, \quad \alpha_\lambda, \beta_\lambda, \gamma_\lambda \in \mathbb{R} \tag{3}$$

Instead of fitting one separate power law function to each learning curve, we fit a single **shared power law function** across all configurations by conditioning the power law coefficients $\alpha, \beta, \gamma$ on $\lambda$ using a parametric neural network $g$ that maps a configuration to the power law coefficients of its learning curve as $g : \Lambda \to \mathbb{R}^3$. The network $g$ has three output nodes, corresponding to the power law coefficients, denoted as $g(\lambda)_\alpha, g(\lambda)_\beta, g(\lambda)_\gamma$, as defined in Equation 4.

$$\hat{f}(\lambda, b) := g(\lambda)_\alpha + g(\lambda)_\beta \ b^{-g(\lambda)_\gamma}, \quad g : \Lambda \to \mathbb{R}^3 \tag{4}$$

Using a history of learning curve evaluations $H^{(N)} := \{(\lambda_i, b_i, f(\lambda_i, b_i))\}_{i=1}^N$ we can train the power law surrogate to minimize the following loss function using stochastic gradient descent:

$$\underset{\hat{f}}{\arg\min} \ \mathbb{E}_{(\lambda, b, f(\lambda, b)) \sim p_{H^{(N)}}} \ \left| f(\lambda_i, b_i) - \hat{f}(\lambda_i, b_i) \right| \tag{5}$$

BO surrogates need to be probabilistic regression models because the acquisition functions require the posterior variance of the predictions. As a result, we train an ensemble of $K$ diverse surrogates $\hat{f}^{(1)}(\lambda, b), \ldots, \hat{f}^{(K)}(\lambda, b)$ by initializing each surrogate with different weights and by training with a different sequence of mini-batches as in prior work [25, 22]. The posterior mean $\mu$ and the posterior variance $\sigma^2$ of the power law ensemble are trivially computed as:

$$\mu_{\hat{f}}(\lambda, b) := \frac{1}{K} \sum_{k=1}^K \hat{f}^{(k)}(\lambda, b), \quad \sigma_{\hat{f}}^2(\lambda, b) := \frac{1}{K} \sum_{k=1}^K \left( \hat{f}^{(k)}(\lambda, b) - \mu_{\hat{f}}(\lambda, b) \right)^2 \tag{6}$$

A commonly used acquisition function in the domain is Expected Improvement (EI) [35] which incorporates both the mean and uncertainty of predictions, applying a trade-off between exploration and exploitation. Consequently, in our work, we use the EI acquisition with the estimated full budget's ($b_{\max}$) posterior mean and variance. We briefly define the acquisition function in Equation 7:

$$\lambda_{N+1} := \underset{\lambda \in \Lambda}{\arg\max} \ \text{EI}\left(\lambda, b_{\max} | H^{(N)}\right), \tag{7}$$

$$\text{EI}(\lambda, b_{\max} | H) := \mathbb{E}_{\hat{f}(\lambda, b_{\max}) \sim \mathcal{N}\left(\mu_{\hat{f}}(\lambda, b_{\max}), \sigma_{\hat{f}}^2(\lambda, b_{\max})\right)} \left[ \max\left\{ \hat{f}(\lambda, b_{\max}) - f(\lambda_{\text{best}}, b_{\max}), 0 \right\} \right]$$

where, $f(\lambda_{\text{best}}, b_{\max})$ corresponds to the best observed loss for any budget $b' \leq b_{\max}$ from the history $H^{(N)}$. After selecting a configuration with our variant of the EI acquisition, we do not naively run it until convergence. Instead, we propose a novel multi-fidelity strategy that advances the selected $\lambda_{N+1}$ of Equation 7 by a small budget of $b_{\text{step}}$, e.g. 1 epoch of training. Therefore, the selected $\lambda_{N+1}$ will be evaluated at $b_{N+1}$ as defined in Equation 8. Notice, our proposed strategy also covers new configurations with no learning curve evaluations in $H^{(N)}$.

$$b_{N+1} := \begin{cases} b_{\text{step}}, & \nexists \lambda_{N+1} : (\lambda_{N+1}, \cdot, \cdot) \in H^{(N)} \\ b_{\text{step}} + \underset{(\lambda_{N+1}, b, \cdot) \in H^{(N)}}{\max} b, & \text{otherwise} \end{cases} \tag{8}$$

We provide the detailed pseudocode of our method at Algorithm 1.

## 5 Experimental Protocol

### 5.1 Benchmarks

**LCBench:** A benchmark that features 2,000 hyperparameter configurations that parametrize the architecture of simple feedforward neural networks, as well as, the training pipeline [51]. The benchmark features 7 numerical hyperparameters and 35 different datasets from the AutoML benchmark [12].

**PD1:** A deep learning benchmark [45] that consists of recent DL (including Transformers) architectures run on large vision datasets such as CIFAR-10, CIFAR-100, ImageNet, as well as statistical modeling corpora and protein sequence datasets from bioinformatics. Every search space includes varying learning curve lengths, ranging from 5 to 1414, and a different number of evaluated hyperparameter configurations ranging from 807 to 2807. The search space includes hyperparameter configurations that parametrize the learning rate, the learning rate scheduler, and the momentum.

**TaskSet:** A benchmark that features different optimization tasks evaluated in 5 different search spaces [34]. For our work, we focus on the Adam8p search space, which is among the largest search spaces

---

**Algorithm 1:** Multi-Fidelity HPO with Deep Power Laws

---

**Input** : Search space $\Lambda$, initial design $H^{(\text{init})}$, budget increment $b_{\text{step}}$
**Output** : Best hyperparameter configuration $\lambda^*$

---

1   Iteration $i \leftarrow 0$; Evaluate initial configurations and budgets $H^{(i)} := H^{(\text{init})}$;

2   **while** *still budget* **do**

3      Fit a DPL ensemble $\hat{f}^{(1)}(\lambda, b), \ldots, \hat{f}^{(K)}(\lambda, b)$ from history $H^{(i)}$ using Equation 5;

4      Recommend the next configuration $\lambda_{i+1}$ and its budget $b_{i+1}$ using Equation 6, 7 and 8;

5      Train $\lambda_{i+1}$ until $b_{i+1}$ and measure the validation loss $f(\lambda_{i+1}, b_{i+1})$;

6      Append to history $H^{i+1} \leftarrow H^i \cup \{(\lambda_{i+1}, b_{i+1}, f(\lambda_{i+1}, b_{i+1}))\}$;

7      $i \leftarrow i + 1$;

8   **end**

9   **return** Best configuration $\lambda^*$ with the smallest validation loss $\min\limits_{(\lambda^*, b, f(\lambda^*, b)) \in H^i} f(\lambda^*, b)$;

---

in the benchmark with 1000 hyperparameter configurations. Every hyperparameter configuration features 8 continuous hyperparameters. The hyperparameters control the learning rate, the learning rate schedulers, and the optimizer. For variety among our benchmarks, we focus on 12 RNN text classification tasks that feature different RNN cells, embedding sizes, and batch sizes.

For a more detailed explanation of the benchmarks, we refer the reader to Appendix F.

## 5.2   Baselines

**Random Search** stochastically samples hyperparameter configurations for the largest possible budget. **Hyperband** uses multiple brackets with different trade-offs of the initial budget and number of epochs to initially train [28]. It then applies Successive Halving (SH) [17] on every bracket. **ASHA** is an asynchronous version of SH [27] that does not wait for all configurations to finish in an SH bracket before starting the next one. Furthermore, **BOHB** is a follow-up of Hyperband that uses TPE [3] to sample the initial hyperparameter configurations of a bracket [9]. **DEHB**, on the other hand, modifies Hyperband by using evolutionary strategies to sample the initial hyperparameter configurations [1]. Similarly, multi-fidelity **SMAC** extends Hyperband but uses random forests to sample the initial hyperparameter configurations for a bracket [30]. We also use the **Dragonfly** Library [19] to compare against BOCA [18], a multi-fidelity method that uses Gaussian Processes to predict the next hyperparameter to evaluate, as well as the fidelity for which it should be evaluated. For all the baselines, we use their official public implementations. We provide additional details in Appendix G.

## 5.3   Architecture & Training

For our method, we use an ensemble of 5 models, where every model consists of a 2-layer feedforward neural network with 128 units per layer and Leaky ReLU for the non-linearity. The architecture of our method is motivated by prior work [25]. Our network has 3 output units, that are then combined with the budget $b$ to yield the power law output. We apply the GLU non-linearity activation only on the $\beta$ and $\gamma$ output units, allowing $\alpha$ to take any value.

We use the L1 loss to train our network, coupled with Adam featuring an initial learning rate of $10^{-3}$. For the first 10 iterations of our multi-fidelity HPO method in Algorithm 1 we train every network of our ensemble for 250 epochs with randomly sampled initial weights. This choice helps the convergence of the weights in the early stage of HPO. Next, we continuously refine the model for 20 epochs every HPO iteration. However, if the optimization stagnates (surrogate fitting loss does not improve) for more than the LC Length + a buffer of $0.2 \times$ LC Length, the training procedure is restarted with random weights, where every model is trained again for 250 epochs and then only refined. During the refining phase, the new data point at an HPO iteration (Line 9 at Algorithm 1) is sampled with repeat on every batch, to learn new data points equally compared to older data points. Since we are working with discrete search spaces, we evaluate the acquisition function exhaustively

on every hyperparameter configuration. When dealing with continuous search spaces, the acquisition function can either be evaluated exhaustively on a discretized version of the search space, or in a gradient-based way. Our implementation of DPL is publicly available.[1]

## 5.4 Protocol

In our experiments, we standardize the hyperparameter values by performing min-max scaling for our method and every baseline. If a baseline has a specific preprocessing protocol, we do not apply min-max scaling but we apply the protocol as suggested by the authors.

The benchmarks do not support a common evaluation metric for configurations (i.e. the function $f$). As a consequence, the evaluation metric for LCBench is the balanced accuracy, for TaskSet the log-likelihood loss, while for PD1 the accuracy. Moreover, the benchmarks do not offer learning curves with a common step size. For LCBench and PD1, one step size is equivalent to one epoch, while for TaskSet one step size is 200 batches. The HPO budget is defined as the maximum number of steps needed to fully evaluate 20 hyperparameter configurations. In that context, one unit step of the HPO budget signifies training a particular configuration for one more optimization step (e.g. 200 batches in TaskSet or 1 epoch in LCBench).

In the following experiments, for all methods, we report the regret of the best-found configuration as shown in Equation 9:

$$R = f\left(\lambda_{\text{best}}, b_{\text{max}}\right) - f\left(\lambda_{\text{oracle}}, b_{\text{max}}\right) \tag{9}$$

where the oracle is given as $f\left(\lambda_{\text{oracle}}, b_{\text{max}}\right) := \min\left\{f\left(\lambda, b\right) \mid \left(\lambda, b, f\left(\lambda, b\right)\right) \in H^{(D)} \wedge b \leq b^{\text{max}}\right\}$, and $H^{(D)}$ corresponds to the set of all the exhaustively-evaluated hyperparameter configurations' performances on a dataset $D$. If the oracle configuration is not known in advance for the search space, $H^{(D)}$ can be replaced with the history $H^{(N)}$ at the end of the HPO procedure. The only difference between $f\left(\lambda_{\text{best}}, b_{\text{max}}\right)$ and $f\left(\lambda_{\text{oracle}}, b_{\text{max}}\right)$ is that the former only considers the history at the HPO step for which we are reporting the results.

In short, the regret is the difference in the evaluation metric performance from the best-found hyperparameter configuration during optimization to the best possible hyperparameter configuration (oracle) on the dataset (in a minimization setting). On a dataset level, we report the average regret across 10 repetitions with different seeds. To be able to aggregate results over datasets, we report the averaged normalized regret. As normalization, we divide the regret by the difference between the performances of the best and the worst hyperparameter configuration on a dataset. Then we compute the mean of the normalized regrets across all the datasets of a benchmark.

Moreover, in the experiments that report the average normalized regret over time, we provide results over normalized wall clock time. The wall clock time includes both the method's overhead (i.e. training the surrogate $\hat{f}$ and selecting the next hyperparameter configuration to evaluate) and the time taken to evaluate the selected hyperparameter configuration (i.e. evaluating $f$). Since the methods have different run times, we normalize the individual times by the time it took Random Search (the fastest non-model-based method) to complete the HPO optimization process. To provide a fair any-time comparison, we report results until the time it took Random Search to evaluate 20 hyperparameter configurations.

Furthermore, when reporting the learning curve (LC) length fraction, we imply the fraction of the total learning curve length. LCBench and TaskSet have LCs of a fixed length for all datasets, corresponding to 51 epochs for LCBench and 50 epochs for TaskSet. In contrast, PD1 has varying LC lengths for different datasets.

In our experiments, all methods start with a history $H^{(init)}$ of 1 randomly sampled hyperparameter configuration evaluated for 1 step/epoch in the case of multi-fidelity techniques (Hyperband, BOHB, DEHB, SMAC, ASHA, Dragonfly; descriptions in Section 5.2), or for the full budget for the black-box technique (Random Search). We ran experiments on a CPU cluster, where every node contains two Intel Xeon E5-2630v4 CPUs with 20 CPU cores running at 2.2 GHz. The total memory of every node is 120GB, and every experiment is limited to 2 cores which offer 12GB.

---

[1] `https://github.com/releaunifreiburg/DPL`

# 6 Research Hypotheses and Experiments

**Hypothesis 1:** *The power law assumption improves the quality of learning curve forecasting.*

In this experiment, we evaluate the predictive performance of forecasting models that given a fraction of the observed learning curve, estimate the remaining unobserved segment of the curve, on the LCBench benchmark. The results of Figure 1 compare three different forecasting models, concretely, neural networks (NN), Gaussian Processes (GP), and Power Law functions (PL). For the three variants (PL, NN, GP) we fitted one model on every learning curve of each hyperparameter configuration (i.e. given $b$ in the x-axis estimate one $\hat{f}(b)$ separately for every $\lambda$). For the other two variants (DPL and Cond NN) we fit a single forecasting model (not an ensemble) for all configurations, by conditioning the

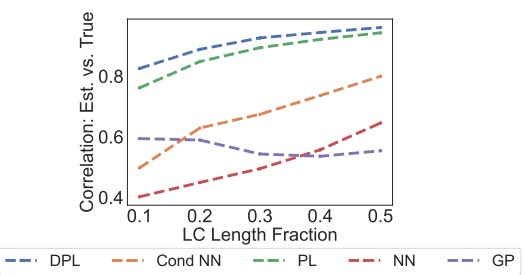

Figure 1: Rank correlations of learning curve forecasting models, which are given a fraction of the learning curve and estimate the remaining curve segment. **DPL**: Deep Power Law, **Cond NN**: Conditioned neural network, **PL**: Power Law, **NN**: Neural Network, **GP**: Gaussian processes.

surrogate on the configuration (i.e. given b and $\lambda$, estimate $\hat{f}(\lambda, b)$). The purpose of the experiment is to assess whether a power law function regressor leads to superior predictive accuracy, compared to generic forecasting models, such as neural networks, or Gaussian processes. The evaluation metric of the experiment highlighted in Figure 1 is the rank correlation between the estimated performances at the end of the learning curve and the true performances.

We notice that although a Power Law regressor has significantly fewer parameters than a neural network (3 to 288 parameters), PL still achieves higher predictive performance than NN. Furthermore, our conditioning of the power law function to the hyperparameter configuration further improves the predictive quality, as the difference between DLP and PL demonstrates.

Lastly, we refer the reader to Appendix H, where we provide an analysis of the distributions for the absolute error rate between the DPL predictions and the ground truth values over the different LC length fractions, showing that DPL does not only retain the ranks but, it also accurately predicts the final performance. Based on the results, we consider Hypothesis 1 to be validated and that **DPL is accurate in terms of learning curve forecasting**.

**What about learning curves that do not follow a power law pattern?** Although the provided empirical evidence in this section strongly suggests that the presented power law model can accurately forecast learning curves, it is also true that some learning curves have divergent behaviour that does not follow the power law assumption. As a consequence, we investigate two different ways to handle curves that do not follow the power law assumption: *i)* recently-proposed power law functions that include breaking points [5], or shifts in the curve [8], and *ii)* min-smoothing the learning curves to transform them into monotonically decreasing time series. In Appendix A we provide ample empirical evidence showing that although the alternative formulations achieve comparable performances, still they do not outperform our simpler power law formulation. The findings indicate that even though not all learning curves are power laws, most of them are, therefore a power law surrogate is efficient in forecasting the final performance of a partially-observed learning curve. As a result, the forthcoming experiments will provide further empirical evidence that our power law surrogates lead to state-of-the-art HPO results when deployed in the proposed multi-fidelity Bayesian optimization setup.

**Hypothesis 2:** *Our method DPL achieves state-of-the-art results in HPO.*

In Figure 2, we show the performance of the considered methods over the HPO budget, where DPL manages to outperform all the rival baselines. In the case of LCBench, DPL quickly finds well-performing hyperparameter configurations compared to the competitor methods and continues to discover even better configurations until the HPO process ends. Furthermore, we observe the same trend with TaskSet and PD1, where after ca. 25% of the HPO budget, our method DPL converges to a better regret compared to the baselines and increases the lead until HPO ends. For a detailed overview of the performances of all methods on all individual datasets, we point the reader to Appendix H.

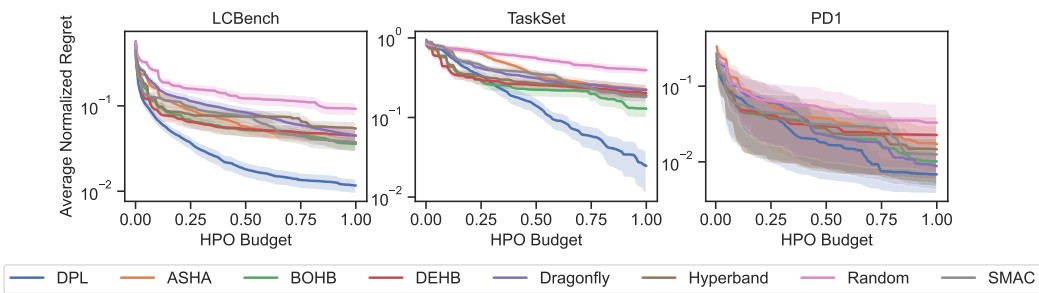

Figure 2: DPL discovers better hyperparameter configurations than all rival baselines in terms of regret (distance to oracle). Solid curves and shaded regions represent the mean and standard error of the averaged normalized regret.

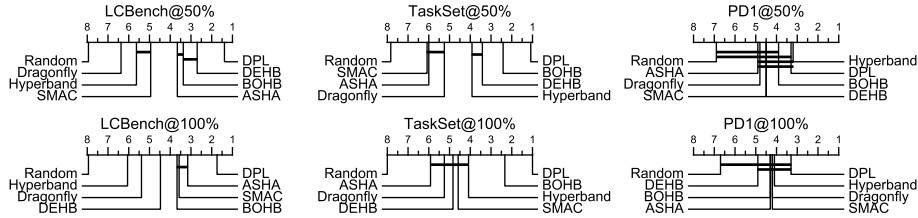

Figure 3: The critical difference diagrams at 50% and 100% of the HPO budget. The ranks indicate the sorted position in terms of regret, aggregated across datasets (the lower the better). Thick horizontal lines highlight differences that are not statistically significant.

In addition, Figure 3 provides the critical difference diagrams of the per-dataset regret ranks at 50% and 100% of the HPO budget. Our method DPL outperforms all baselines in 5 out of 6 cases (in 4 of which with a statistically significant margin), while being second best only at the 50% of the HPO budget on the PD1 benchmark. We investigate the lack of statistical significance in PD1, by analyzing the individual dataset performances where DPL performs worse compared to other baselines. We notice that the datasets have a skewed distribution of hyperparameter configuration performances, where, a majority of the configurations achieve top performance. Based on the results, we conclude that a lack of statistical significance is the outcome of a search space that includes relatively simple optimization tasks and not a specific failure state of our method. We provide the detailed results of our analysis in Appendix D.

Lastly, we analyse the performance of DPL over time in Figure 4. As it can be observed, DPL manages to outperform the competitors even when the method's overhead time is included, showing that the overhead of DPL (i.e. fitting

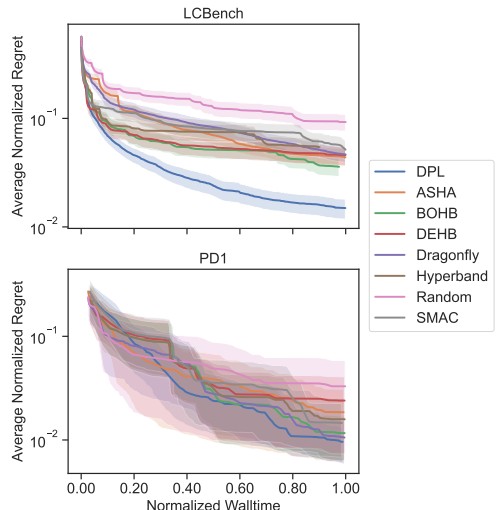

Figure 4: The average normalized regret of DPL and rival methods over the normalized time for all the considered benchmarks. Solid curves and shaded regions represent the mean and standard average normalized regret.

surrogate and running the acquisition) is negligible in terms of the quality of the HPO results. For a more detailed information, regarding the DPL overhead time, we point to Appendix E. TaskSet is not included in Figure 4 since the benchmark does not offer runtimes. Given the results, we conclude that Hypothesis 2 is validated and that **DPL achieves state-of-the-art performance in HPO**.

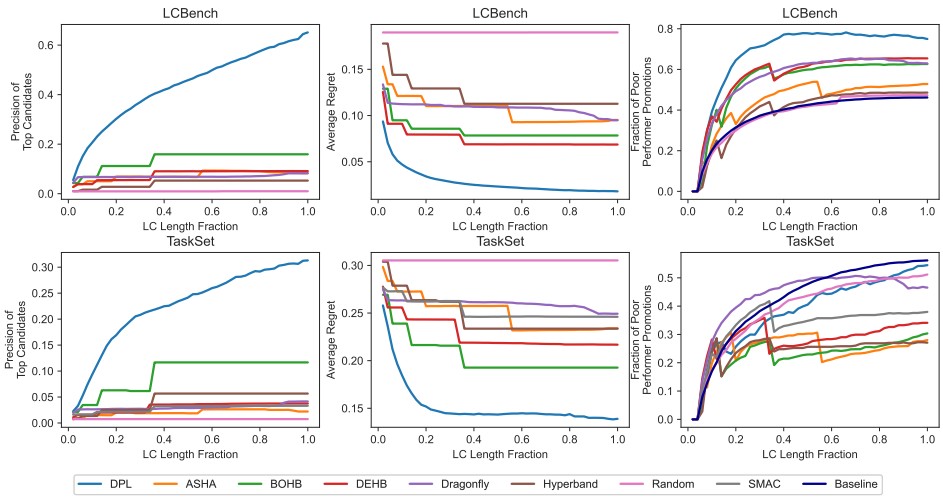

Figure 5: Post-hoc analysis to study DPL's efficiency. **Left:** Share of the best candidates selected during training. **Middle:** Average regret of configurations chosen to be trained at each budget. **Right:** Share of top third configurations at a given budget which were bottom two third configurations at a previous budget.

**Hypothesis 3:** *DPL explores the search space more efficiently compared to the baselines.*

We conduct further analyses to understand the source of the efficiency of DPL versus the baselines. As a result, we analyze two important aspects, the quality of the evaluated configurations, as well as the exploration capability of our multi-fidelity HPO method. Initially, we measure what fraction of the top 1% configurations (ranked by accuracy) can our method discover. Figure 5 (left) shows that until convergence our method can discover significantly more top configurations compared to the baselines. The middle plots of Figure 5, show the average regret for each configuration promoted to the respective budget. According to the plot, DPL is more efficient and assigns the budget only to configurations with lower regret compared to the other methods. The precision and regret plots demonstrate that the quality of the evaluated configurations is largely better than all baselines, therefore, giving our method a significant lift in the performance rank. Last but not least, the right plot shows the percentage of configurations that were performing poorly in an earlier epoch (i.e. accuracy-wise in the bottom $2/3$ of configurations up to the epoch indicated at the x-axis) but performed better at later epochs (i.e. at the top $1/3$ of configurations). Furthermore, we added a line labeled with "Baseline", which represents the fraction of previously poor-performing configurations of all configurations. This behavior is observed often with learning curves, where for instance, strongly regularized networks converge slowly. For the same analysis regarding the PD1 benchmark, we point the reader to Appendix H.

The results indicate that our method explores well the unpromising early configurations, by considering them through the uncertainty estimation of our ensemble and the respective Bayesian optimization mechanism. The results validate Hypothesis 3 and confirm that **DPL explores the search space more efficiently.**

**Hypothesis 4:** *Our method DPL offers an effective tool for HPO in Large Language Models.*

In this experiment, we consider the case of tuning the hyperparameters of transformers in Large Language Models. To this end, we computed a tabular benchmark by training a smaller GPT-2 [39] model on the OpenWebText dataset [13] for a series of different hyperparameter configurations. We tune three learning rate hyperparameters: the fraction of warmup steps, the maximum learning rate at the end of warmup, and the minimum learning rate at the end of the decay. We repeat the experiments for seven model sizes ranging from 0.3M to 30M total parameters, ablating the embedding size of the multi-head attention layers (details in Appendix B).

We follow the common practice of conducting HPO with small transformers and then deploying the discovered optimal configuration on the full-scale transformers [50]. As a result, we search for the optimal hyperparameters of small transformers (embedding size of $\{6, 12, \ldots, 96, 192\}$) and then evaluate the discovered configurations at a full-scale transformer with an embedding size of $384$.

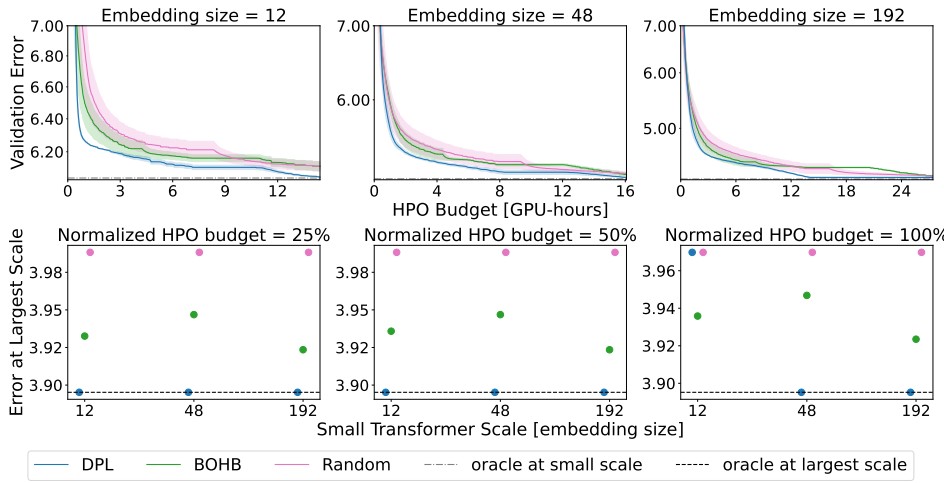

Figure 6: HPO for Transformer architectures. **Top:** HPO on small-scale transformers in terms of the embedding size. **Bottom:** Error on the full-scale transformer, using the hyperparameter configuration discovered by conducting HPO using the small transformers. We present three analyses, ablating the HPO time on the small-scale transformer up to the HPO budget of 2 full function evaluations.

Figure 6 shows the HPO results of DPL against Random Search and BOHB (a rival multi-fidelity HPO baseline). In the top row of plots, we observe the performance of the discovered configurations at the small transformers for the indicated embedding size. We observe that our method finds better configurations than the baselines at any proxy space with small embedding sizes.

On the other hand, the bottom row of plots presents the performance of the discovered configurations in the small embedding space, by applying such hyperparameter configurations to the full-scale transformers. We observe that the configurations discovered by DPL on the small search space achieve very competitive results on the full-scale transformers, finding the oracle configuration of the full-scale transformers in the majority of cases. It takes DPL 3.6 hours to find the oracle configuration for the largest model via HPO for the smallest model. In turn, it takes 21.52 hours to train the largest model only once. For more details, we refer the reader to Appendix B. The results validate Hypothesis 4 and confirm that **DPL is an efficient HPO technique for tuning the hyperparameters of large language models when the HPO is conducted using smaller transformer model sizes.**

## 7   Conclusions

**Summary.** In this work, we introduce Deep Power Law (DPL), a probabilistic surrogate based on an ensemble of power law functions. The proposed surrogate is used within a novel multi-fidelity Hyperparameter Optimization (HPO) method based on Bayesian optimization. In contrast to the prior work, we exploit scaling laws for estimating the performance of Deep Learning (DL) models. Through extensive experiments comprising 7 baselines, 59 datasets, and search spaces of diverse deep learning architectures, we show that DPL outperforms strong HPO baselines for DL by a large margin. As an overarching contribution, we advance the state-of-the-art in the important field of HPO for DL.

**Limitations and future work.** Contrary to the common perception, we experienced that the uncertainty estimation arising from the Deep Ensemble approach [25] is suboptimal compared to standard BO surrogates such as Gaussian Processes. In addition, having to train an ensemble has additional computational costs, due to the necessity of training multiple power law models. In the future, we plan to investigate the combination of power laws with Gaussian Processes, as well as investigate additional fidelity types.

## Acknowledgements

**JG**, **AK** and **MJ** would like to acknowledge the grant awarded by the Eva-Mayr-Stihl Stiftung. In addition, this research was funded by the Deutsche Forschungsgemeinschaft (DFG, German Research Foundation) under grant number 417962828 and grant INST 39/963-1 FUGG (bwForCluster NEMO). In addition, **JG** and **AK** acknowledge the support of the BrainLinks-BrainTools center of excellence.

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

# A   Modeling

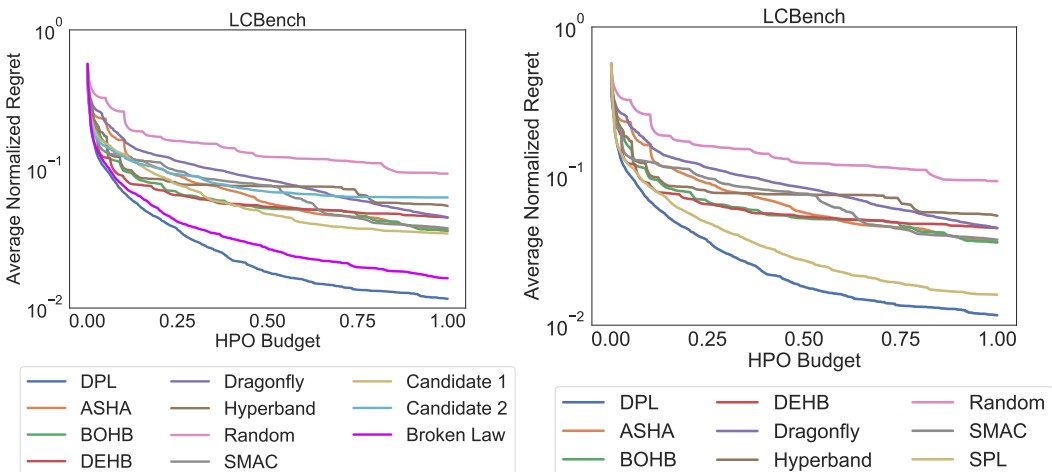

Figure 7: **Left:** The performance of different power law formulations, as well as, the baselines on the LCBench benchmark over the HPO budget. **Right:** The performance of the power law formulation when min-smoothing is applied to the learning curve to ensure a monotonically decreasing learning curve.

To inspect the modeling choices of the power law functions used as our surrogate, we consider different formulations for the ensemble members of our surrogate as shown in Table 1. Initially, we consider Candidate 1 which can handle shifts in the learning curve by introducing an additional parameter $d$. Furthermore, we consider a more complex version, Candidate 2, that allows us to additionally scale the budget, by introducing variable $e$. Lastly, we consider

| Label | Formula |
|---|---|
| DPL | $\alpha + \beta \cdot b^{-\gamma}$ |
| Candidate 1 | $\alpha - \beta \cdot (b + d)^{-\gamma}$ |
| Candidate 2 | $\alpha - \beta \cdot (e \cdot b + d)^{-\gamma}$ |
| Broken Law | $\alpha + \beta \cdot b^{-\gamma} \cdot \left(1 + \left(\frac{b}{d}^{\frac{1}{f}}\right)\right)^{-c \cdot f}$ |

Table 1: Alternative power law formulations for the DPL surrogate.

Broken Laws [5] (BPL), which can handle breaking points in the learning curve. We use a version that can handle one breaking point since the authors of the method suggest it as a sufficient formulation to approximate the majority of cases. We run the DPL surrogate with every individual formulation on the LCBench benchmark to investigate their empirical performance.

Figure 7 presents the results, where, our chosen surrogate formulation (DPL) despite being the simplest, outperforms all other formulations. The Candidate 2 formulation does not manage to outperform all competitor methods, the Candidate 1 formulation manages to outperform all rival methods, however, only with a marginal difference. The Broken Law formulation manages to outperform all the rival baselines by a considerable margin, however, it still performs worse than the simple power law formulation used for DPL.

We would like to point out that the alternative power law formulations are more difficult to optimize/run since they are prone to diverge and fall into a dead state given different combinations of parameter values. The most common is division by zero for e.g. $d$ term in the Broken Law formulation, taking the root of a negative number $b + d$ term in Candidate 1, $e \cdot b + d$ for Candidate 2 and the $d$ term in Broken Law.

We additionally consider min-smoothing the learning curve $y$, by taking at each step $b$ of the learning curve the minimal value of the learning curve $y_b^{smooth}$ where, $y_b^{smooth} = \min{(y_0, y_1, ..., y_b)}$. The min-smoothing allows diverging curves to be formulated as power laws since the diverging phase will be substituted with stagnation. Figure 7 shows that incorporating learning curve min-smoothing for our surrogate (SPL) performs comparably to DPL without learning curve smoothing and manages to beat the other HPO baselines.

# B  nanoGPT-Bench

In recent years, deep learning research has increasingly focused on large-scale models, particularly Large Language Models (LLMs) like the Generative Pre-trained Transformers (GPTs). To evaluate the effectiveness of search algorithms, we propose a benchmark based on the nanoGPT model [21], reproducing the performance of the small GPT-2 [39] model trained on the OpenWebText dataset [13].

Our experimental setup is designed with the practicalities of real-world hardware constraints in mind. The common practice in the field is to perform Hyperparameter Optimization (HPO) on an informative proxy task that adheres to model size scaling laws [37], and then to apply these optimized parameters to larger models. This pragmatic approach is necessitated by the fact that training larger models would require significantly more expensive computational resources and time. Within these constraints, we focus our experiments on NVIDIA RTX 2080 GPUs.

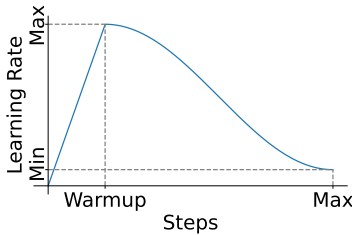

Figure 8: nanoGPT-Bench search space parametrization.

**Baseline:**  We train a small nanoGPT model, a scaled-down variant of the small GPT-2, reducing the parameter count to approximately $30$ M from the original $124$ M parameters. The model architecture includes 6 transformer layers and 6 attention heads, and the embedding size is set to $384$. The AdamW optimizer is utilized for training, with the first and second moment estimates configured to $0.9$ and $0.98$, respectively. The weight decay is set to $10^{-1}$, and we apply gradient clipping at a value of $1.0$ to prevent large gradients from causing instability in the model training. For the training process, we optimize the cross-entropy loss for next-token prediction. The process involves $350$ steps, with each step encompassing $1000$ random samples, with a batch size of $12$. Each data point has a context size of $512$ tokens, encoded using OpenAI's token embeddings (sized of $50304$). This procedure ensures that even the most resource-intensive experiments stay within the limits of a single GPU day.

**Search Space:**  Our hyperparameter search space construction involves varying the number of warmup steps, along with the maximum and minimum learning rates for the cosine scheduler. The specific parametrization of the scheduler is illustrated in Figure 8. The discretized choices are presented in detail in Table 2.

| HP | Values |
|---|---|
| Max LR | $[10^{-5}; 10^{-4}; 10^{-3}]$ |
| Min LR | $[1\%; 10\%]$ of Max LR |
| Warmup Steps | $[10\%; 20\%]$ of Budget |

Table 2:  Search space of nanoGPT-Bench.

**Fidelity Space:**  To construct the fidelity space, we focus on two key dimensions: the number of training steps and the transformer's embedding size, serving as a proxy for model size. In this exploration, the natural fidelity of the number of training steps is visualized by the validation curves during model training as depicted in Figure 9. On the other hand, the end performance correlation between the different fidelities is reflected by the Pearson correlation in Table 3. We establish proxy tasks by sampling embedding size from a log scale, $\{6, 12, \ldots, 96, 192\}$, with the maximum being $384$. Consequently, each configuration has 6 proxy and one target task, leading to a total of $84$ unique configurations in our full benchmark. Every configuration is trained for 350 steps.

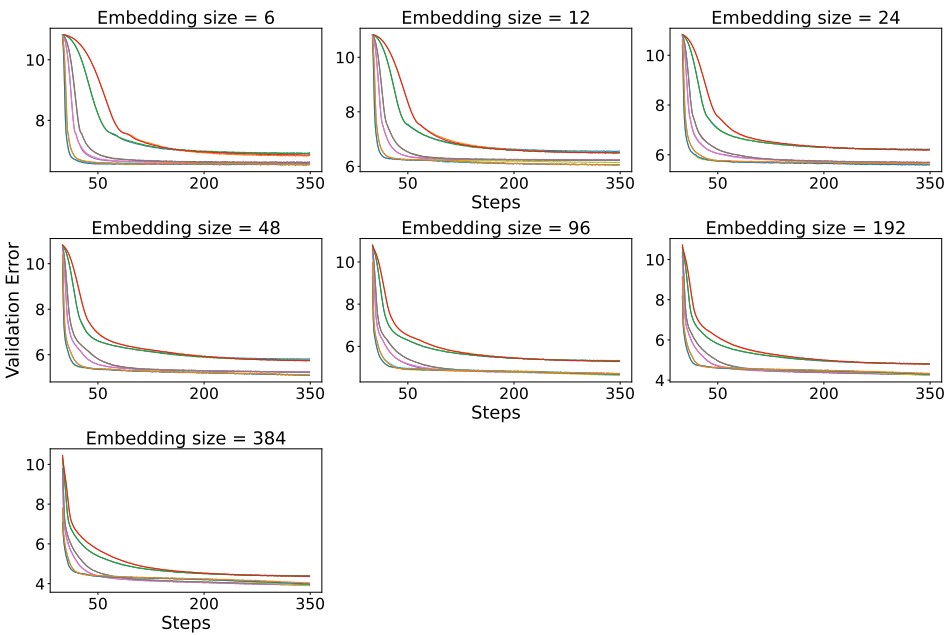

Figure 9: Validation loss curves during model training for all nanoGPT-Bench configurations across model fidelity values.

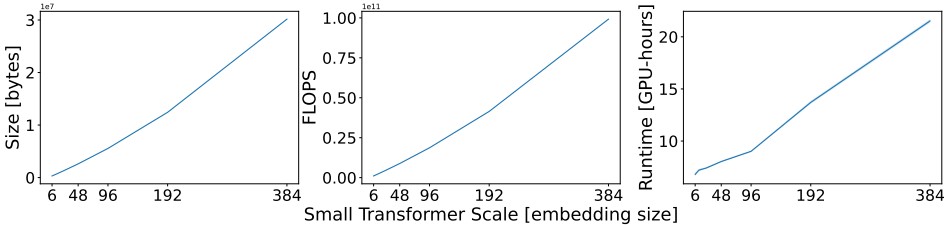

Figure 10: Scaling of model size in relation to bytes, FLOPS, and runtime based on average values across all nanoGPT-Bench configurations.

Due to the GPU under-utilization of small model sizes, the runtime scales linearly as the model size scales exponentially. This relationship can be observed in Figure 10 and Figure 11. We expect the runtime to scale in a linear proportion to the model size when larger models are considered.

Figure 12 illustrates the effectiveness of DPL, particularly when the number of training steps is considered as a fidelity dimension. Vertical dotted lines denote the iteration at which an algorithm identifies the oracle value with an absolute tolerance of $0.01$.

Figure 13 depicts the results over the different values of the embedding fidelity. We utilized DPL, BOHB, and random search in proxy tasks, incrementing the budget allocation over each run, up to a horizon of 6 full-function evaluations. From these proxy tasks, we extracted the incumbent hyperparameters and evaluated their performance on the target task, that correponds to the maximum embedding size of $384$.

| Embedding Size | Correlation |
|:--------------:|:-----------:|
| 6 | 0.951 |
| 12 | 0.880 |
| 24 | 0.971 |
| 48 | 0.955 |
| 96 | 0.987 |
| 192 | 0.994 |
| 384 | 1.000 |

Table 3: Pearson correlation across 7 fidelities.

Despite operating within short regimes, DPL consistently outperforms baselines in terms of the mean incumbent value and the explored regime, as evidenced by error bars indicating the range between best and worst incumbents across 10 seeds. It should be noted, however, that the correlation between

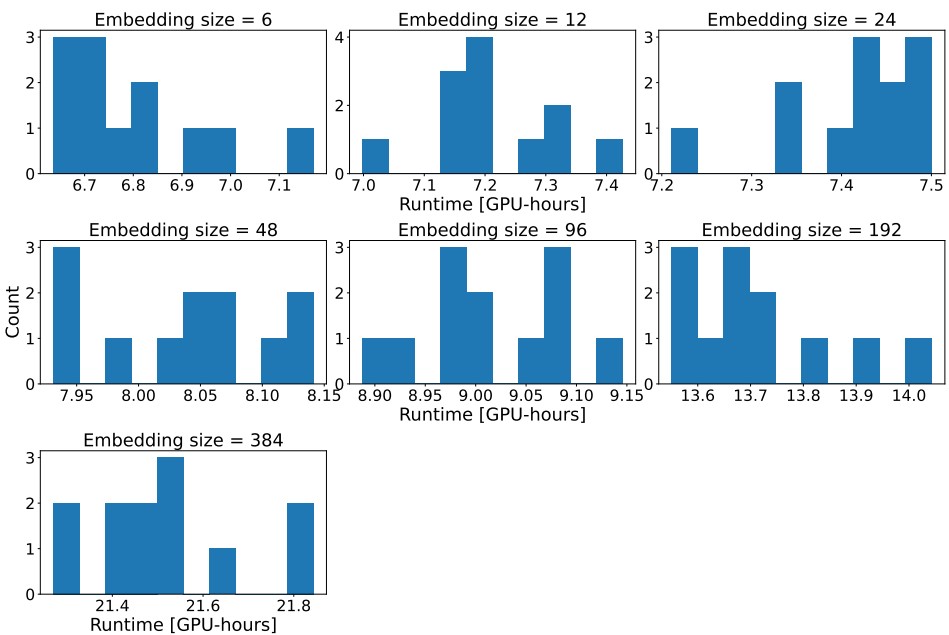

Figure 11: Distribution of GPU-hours required for training across different model fidelity values.

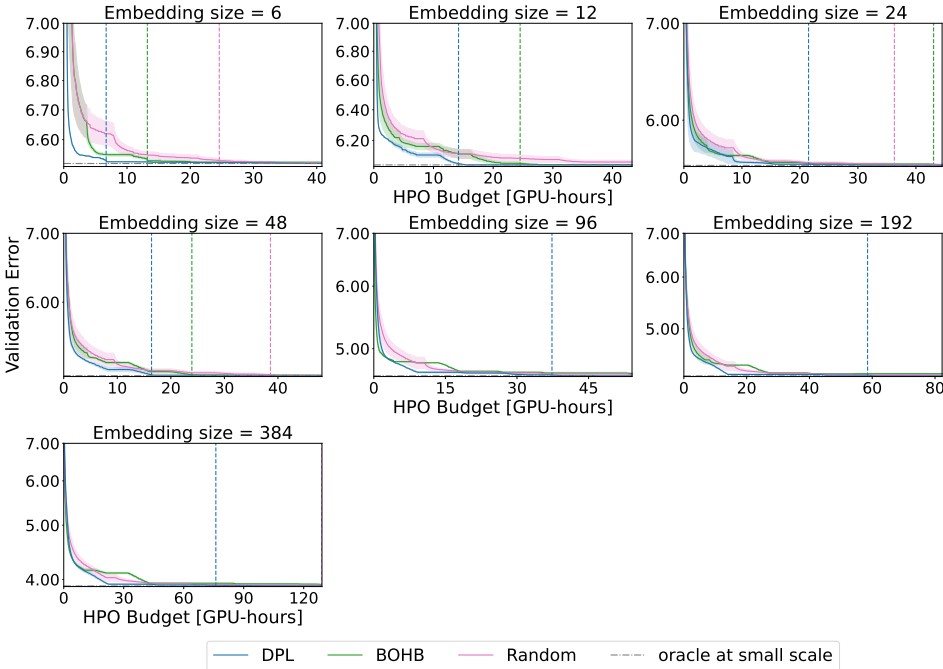

Figure 12: The incumbent performance of DPL and other baselines during the HPO budget of 6 full function evaluations for different values of the embedding size fidelity. Dashed lines indicate the point at which the oracle has been evaluated for every algorithm. Solid curves and shaded areas stand for mean value across runs and standard error.

proxy and target tasks is not always perfect. This can result in a proxy task incumbent that does not translate to the oracle in the target task, which can be observed particularly at lower fidelity levels.

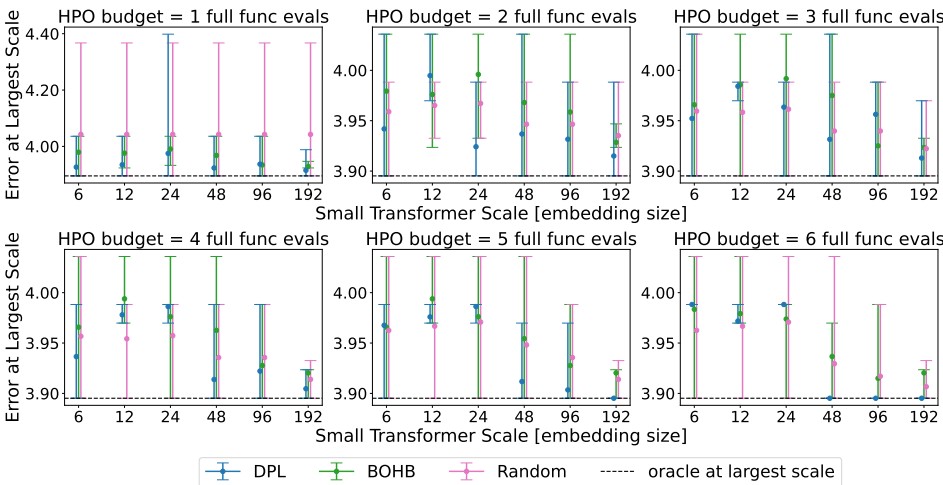

Figure 13: The target task performance distribution for DPL and other methods over different HPO budgets ranging from $1 - 6$ full function evaluations.

## C  Continuous Search Space

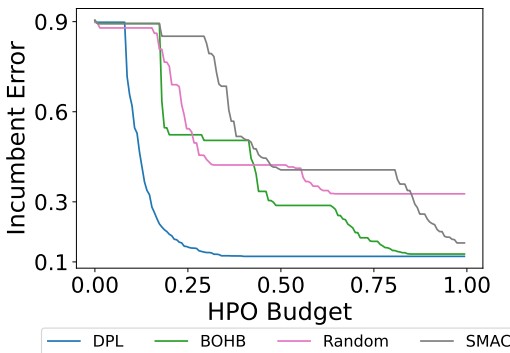

Figure 14: The incumbent error of DPL, as well as, the baselines on the CIFAR10 task over the HPO budget.

The primary objective of this study is to investigate the efficacy of Deep Power Laws (DPL) in trading off exploration vs exploitation in a continuous HPO search space. In this study, we do not make use of pre-computed tabular tables, but instead we optimize the hyperparameters of an EfficientNetV2 model online, by iteratively pausing unpromising configurations and moving forward only promising hyperparameter configurations during the HPO optimization procedure.

To benchmark our findings, we contrast the results against established baseline algorithms such as random search, BOHB, and SMAC.

**Baseline:**  We employ EfficientNetV2 [43] as a benchmarking model and train it on the CIFAR10 dataset [24]. Specifically, we train the lightweight variant of EfficientNetV2-b0 from scratch for $50$ epochs, using the RMSprop optimizer. The learning rate is initiated at $10^{-6}$ and gradually increased over a span of five warmup epochs to reach the learning rate value of $5 \cdot 10^{-4}$. Following the warmup phase, we employ a cosine learning rate scheduler, with a decay factor of $0.97$ applied every $10$ epochs. The weight decay is set at $10^{-5}$, with no momentum used. Furthermore, the dropout rate is configured to be $10^{-6}$ and the model's moving average exponential decay is set at $0.9996$. During the training phase, the batch size is set to $64$, while for the validation phase, it is reduced to $8$. All experiments are performed using the timm library [46].

**Search Space:**    In our experiment, we concentrate on optimizing the two most critical hyperparameters, learning rate, and weight decay, while keeping the remaining hyperparameters fixed as per the baseline model. We construct a search space for these two hyperparameters in accordance with common practices (Table 4).

| HP | Values |
|---|---|
| LR | $[10^{-5}, 10^{-2}]$ |
| weight decay | $[0, 10^{-1}]$ |

Table 4:  Search space of CIFAR10 task.

To emulate a continuous search space for the acquisition function, we generate 100 equally-sized steps on a logarithmic scale from the lower bound to the upper bound of each dimension. This process yields a search space comprising $10^4$ potential configurations.

Our method demonstrates a substantial speedup in terms of anytime performance when compared to baseline algorithms (Figure 14). Acknowledging the practical constraints of evaluating large-scale models, we pragmatically allocated an HPO budget for a maximum of 3 full-function evaluations, equivalent to 150 epochs. The results of our exploration underline the compelling potential of the DPL algorithm to effectively manage HPO tasks within a continuous search space. Exhibiting significant speedup gains, DPL proves itself to be not only viable but an efficient method for identifying optimal hyperparameters. The findings further underscore the adaptability and efficacy of DPL in addressing complex HPO tasks, reinforcing its standing as a valuable tool in the machine learning toolbox.

# D    PD1 Investigation

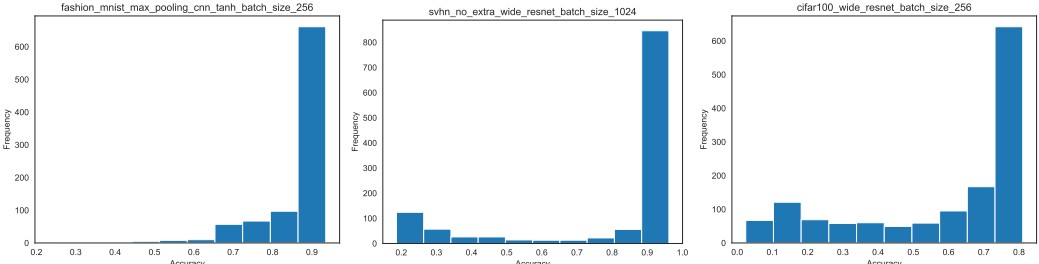

Figure 15: Configuration test performance histograms for datasets where DPL does not outperform baselines.

We investigate the datasets: $fashion\_mnist\_max\_pooling\_cnn\_tanh\_batch\_size\_256$, $cifar100\_wide\_resnet\_batch\_size\_256$, $svhn\_no\_extra\_wide\_resnet\_batch\_size\_1024$, where DPL does not outperform other methods for the PD1 benchmark as shown in Figure 23. We analyze the test performance of the individual hyperparameter configurations that belong to the aforementioned datasets. Figure 15 shows that the search spaces of the datasets have a skewed distribution of performances, where, there exist a large number of hyperparameter configurations that achieve top performance. In such datasets, even a non-model based technique will quickly find a well-performing configuration, since, there is a high chance for a randomly-sampled configuration to achieve the top performance. For this reason, there is little room for sophisticated HPO techniques in these datasets.

To further validate our hypothesis, we investigate two additional datasets $dionis$ from the LCBench benchmark and $uniref50\_transformer\_batch\_size\_128$ from the PD1 benchmark, where DPL achieves a strong performance compared to baseline HPO methods. The results in Figure 16 show that DPL excels on tasks that are complex and that require optimizations to find hyperparameter configurations that achieve top performance.

Based on the results, we conclude that the lack of statistical significance in PD1 is not a specific failure mode of DPL, but a consequence of multiple PD1 datasets where the majority of configurations achieve the top performance.

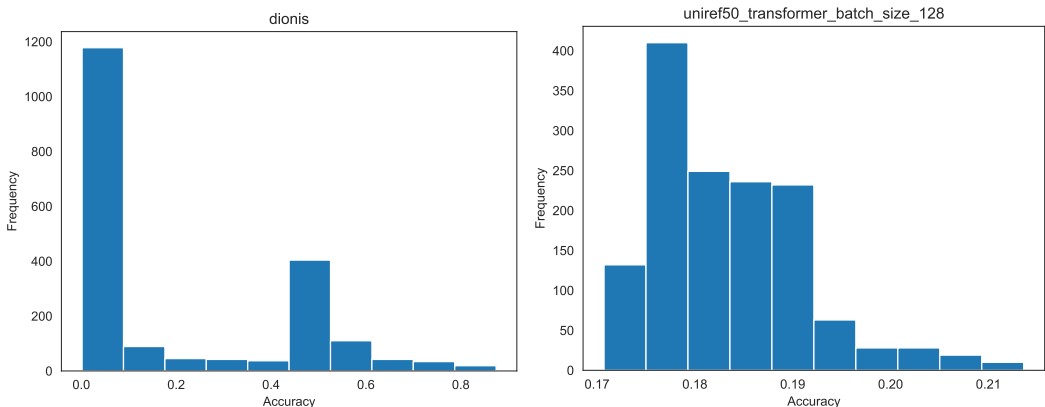

Figure 16: Histograms of the distribution of performances for datasets where DPL performs strongly.

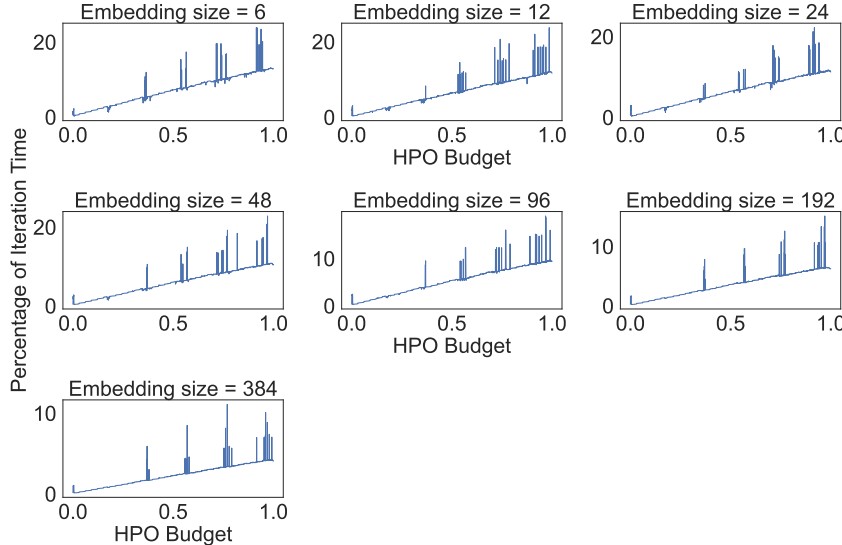

Figure 17: The percentage taken by the DPL overhead in the total time per iteration for the different embedding sizes in nanoGPT-Bench.

# E   DPL Overhead

To investigate the efficiency of DPL in terms of method runtime, we investigate the percentage of time that the DPL overhead contributes in the total time taken to perform one HPO iteration in the nanoGPT-Bench. The total time taken constitutes of the DPL overhead (training the DPL surrogate, calling the acquisition function to suggest the next hyperparameter configuration) and the time taken to run the target algorithm for one more step. Figure 17 shows that the impact of the DPL overhead is negligible in the total time taken. For the smallest embedding of size 6, DPL takes only 10% of the total time taken to perform one HPO iteration after spending half of the optimization budget. At the end, after circa 40 hours of HPO optimization, DPL has an impact of 20% in the total time taken to perform one HPO iteration. The impact is even smaller for the largest embedding of size 384, where DPL has an impact of only 5% in the total time taken per iteration after spending half of the optimization budget and it has an **impact of only 10% in the total time per iteration after more than 120 hours of HPO optimization**.

The findings validate our claim that DPL has a minor time overhead in performing hyperparameter optimization, which explains the strong any-time performance of our method.

# F  Details of Considered Benchmarks

**LCBench:**  We use the official implementation as the interface for the LCBench benchmark [2]. As suggested by the authors, we use the benchmark information starting from the second step and we skip the last step of the curve since it is a repeat of the preceding step.

**TaskSet:**  The TaskSet benchmark features 1000 diverse tasks. We decide to focus on only 12 NLP tasks from the TaskSet benchmark to add variety to our entire collection of datasets. Our limitation on the number of included tasks is related to the limited compute power, as we are unable to run for the entire suite of tasks offered in TaskSet. TaskSet features a set of 8 hyperparameters, that consists of i) optimizer-specific hyperparameters, such as the learning rate, the exponential decay rate, $\beta_1$ and $\beta_2$, and Adam's constant for numerical stability $\varepsilon$, ii) hyperparameters that control the linear and exponential decay schedulers for the learning rate decay, and lastly iii) hyperparameters that control the L1 and L2 regularization terms. Every hyperparameter in TaskSet except $\beta_1$ and $\beta_2$ is sampled logarithmically.

**PD1:**  We use the synetune library [42] for our interface to the PD1 benchmark. From the benchmark, we only include datasets that have a learning curve of length greater than 10. We furthermore only include datasets that have a learning curve lower or equal to 50 to have a fair comparison between all benchmarks by having approximately 20 full-function evaluations. PD1 features 4 numerical hyperparameters, $lr\_initial\_value$, $lr\_power$, $lr\_decay\_steps\_factor$ and $one\_minus\_momentum$, where $lr\_initial\_value$ and $one\_minus\_momentum$ are log sampled. The learning rate decay is applied based on a polynomial schedule, its hyperparameters taken from the search space.

# G  Baselines

**Random Search:** We implemented random search by randomly sampling hyperparameter configurations from the benchmarks with the maximal budget.

**Hyperband, BOHB, LCNet:** We use version 0.7.4 of the HpBandSter library as a common codebase for all 3 baselines [3]. For the last approach mentioned, despite heavy hyperparameter tuning of the method, we could not get stable results across all the benchmarks and hence dropped the method from our comparison.

**ASHA:** For the implementation of ASHA we use the public implementation from the optuna library, version 2.10.0.

**DEHB:** We use the public implementation offered by the authors [4].

**MF-DNN:** In our experiments we used the official implementation from the authors [5]. However, the method crashes which does not allow for full results on all benchmarks.

**SMAC:** For our experiment with SMAC we used the official code base from the authors [6].

**Dragonfly:** We use version 0.1.6 of the publicly available Dragonfly library.

For all the multi-fidelity methods considered in the experiments, we use the same minimal and maximal fidelities. In more detail, for the LCBench, TaskSet and PD1 benchmarks we use a minimal fidelity lower bound of 1 and a maximal fidelity lower bound equal to the max budget.

# H  Plots

In Hypothesis 1, we prove that DPL achieves a better performance in comparison to other models in estimating the final performance for different hyperparameter configurations based on partial observations. Our analysis shows that DPL manages to retain the ranks of different hyperparameter

---

[2]`https://github.com/automl/LCBench`
[3]`https://github.com/automl/HpBandSter`
[4]`https://github.com/automl/DEHB/`
[5]`https://github.com/shib0li/DNN-MFBO`
[6]`https://github.com/automl/SMAC3`

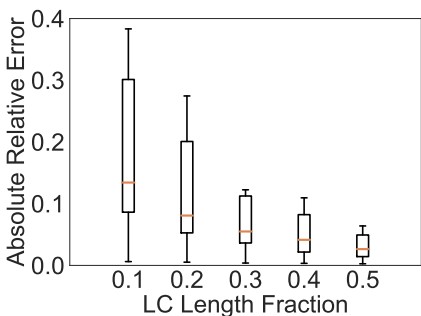

Figure 18: The absolute relative error distribution of DPL over the different learning curve fractions in the LCBench benchmark. The distribution is calculated from the ground truth and prediction values, averaged over all configurations of a dataset. Every distribution is over the datasets.

configurations. We complement Hypothesis 1 by additionally investigating the absolute relative error. We measure the difference between the DPL estimation of the final learning curve value for different fractions of available partial observations vs the actual end performance. Figure 18 shows that DPL does not only retain the ranks of the final performances of different hyperparameter configurations, but it also correctly estimates the final performance by attaining a small relative error, where the error is reduced the more partial observations we have from the learning curve.

In Hypothesis 3, we investigate the efficiency of DPL in exploring more promising configurations compared to other HPO methods. In Figure 19 we provide the same comparison with regards to the PD1 benchmark. Based on the results, we conclude that DPL explores more promising configurations compared to other HPO methods.

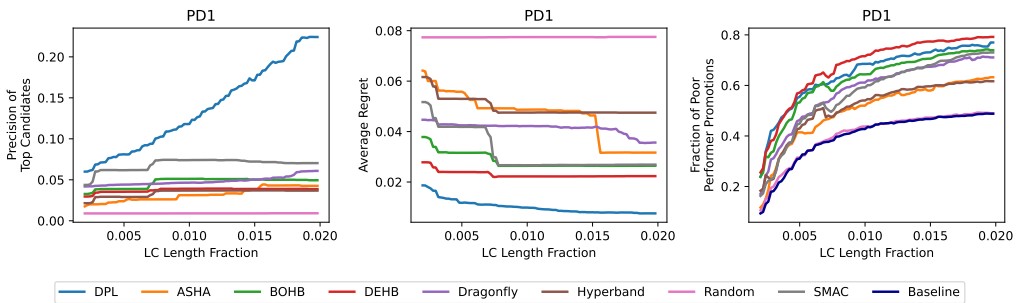

Figure 19: Post-hoc analysis to study DPL's efficiency. **Left:** Share of the best candidates selected during training. **Middle:** Average regret of configurations chosen to be trained at each budget. **Right:** Share of top third configurations at a given budget which were bottom two third configurations at a previous budget.

Lastly, we provide the per-dataset performances of all methods, where we present the mean regret of the incumbent trajectory and the standard error over 10 runs in LCBench (Figure 20 and 21), TaskSet (Figure 22), and PD1 (Figure 23). The results show that DPL consistently outperforms other methods in the majority of cases achieving strong any-time performance and not only a strong final performance.

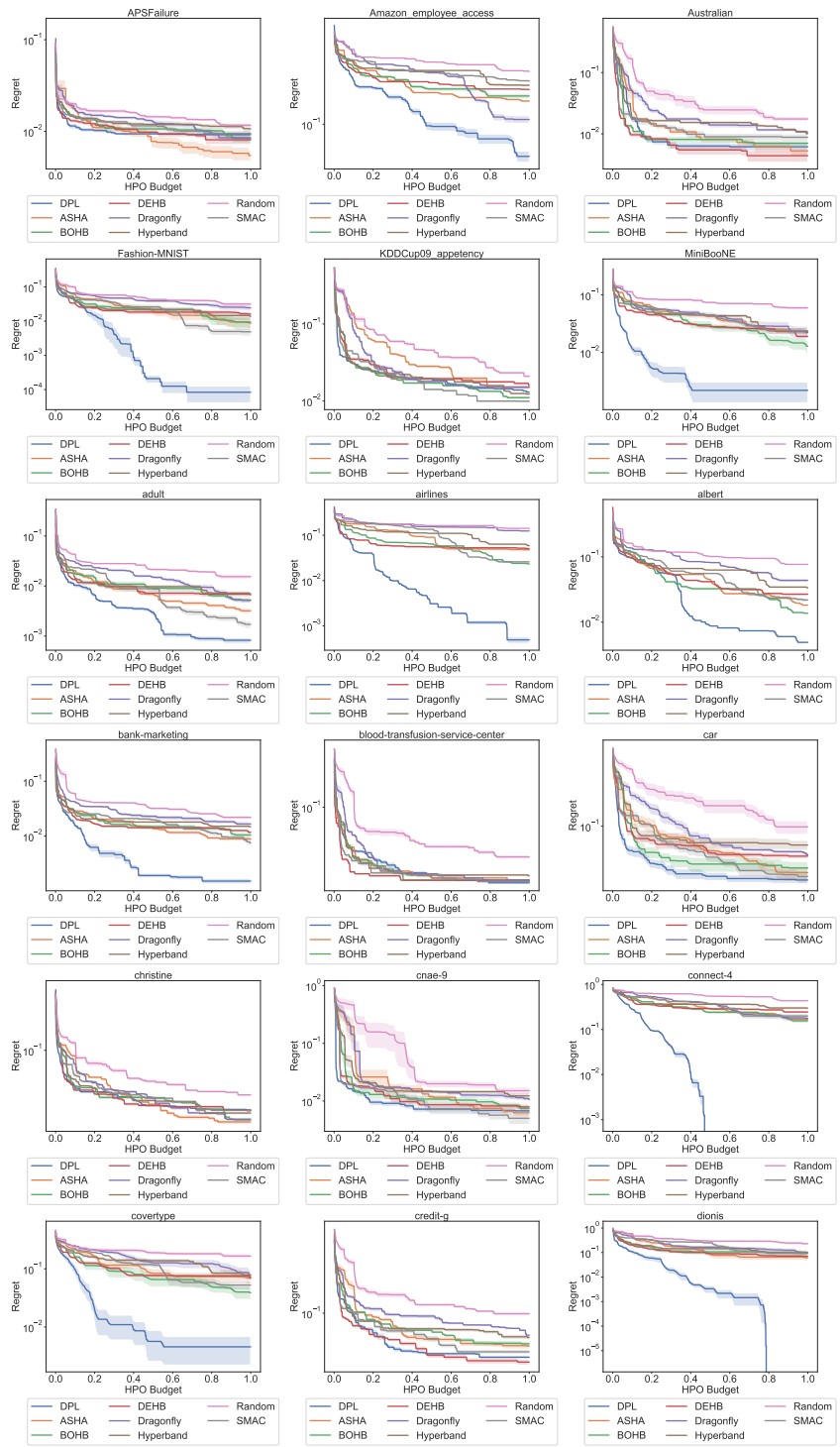

Figure 20: Performance comparison over the number of epochs on a dataset level for LCBench. We plot the mean value over 10 runs and the standard error.

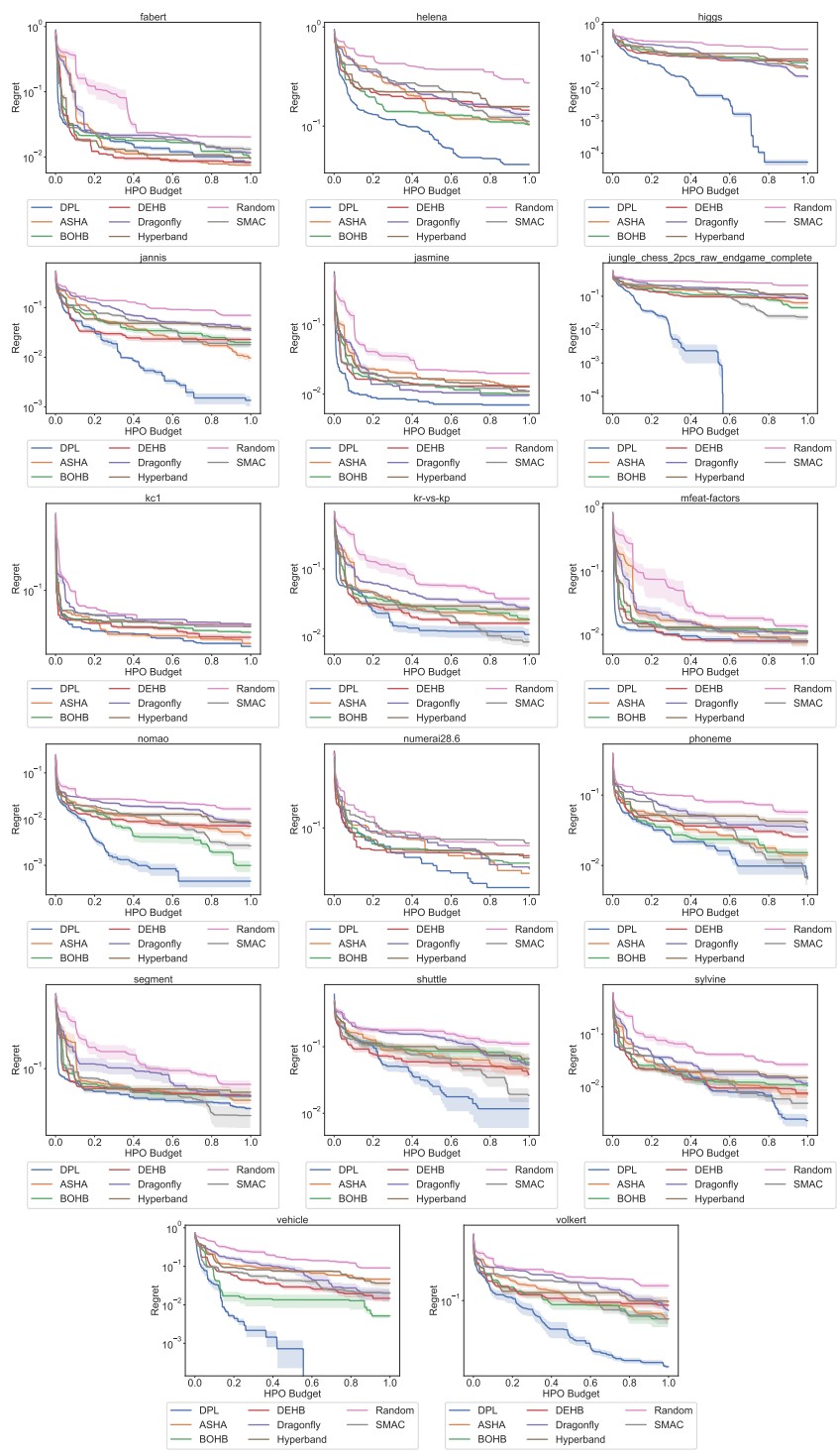

Figure 21: Performance comparison over the number of epochs on a dataset level for LCBench (cont.). We plot the mean value over 10 runs and the standard error.

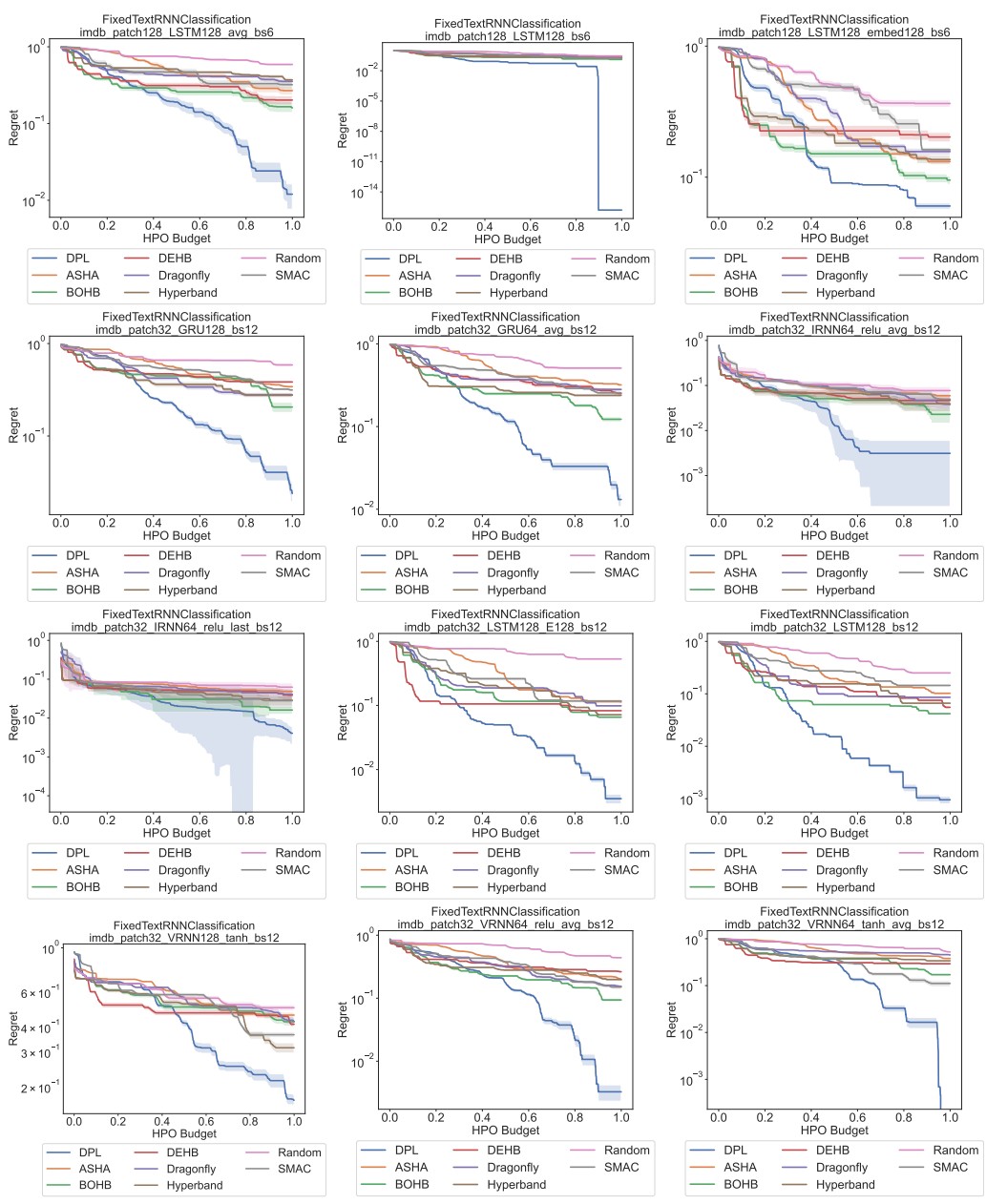

Figure 22: Performance comparison over the number of steps on a dataset level for TaskSet. We plot the mean value over 10 runs and the standard error.

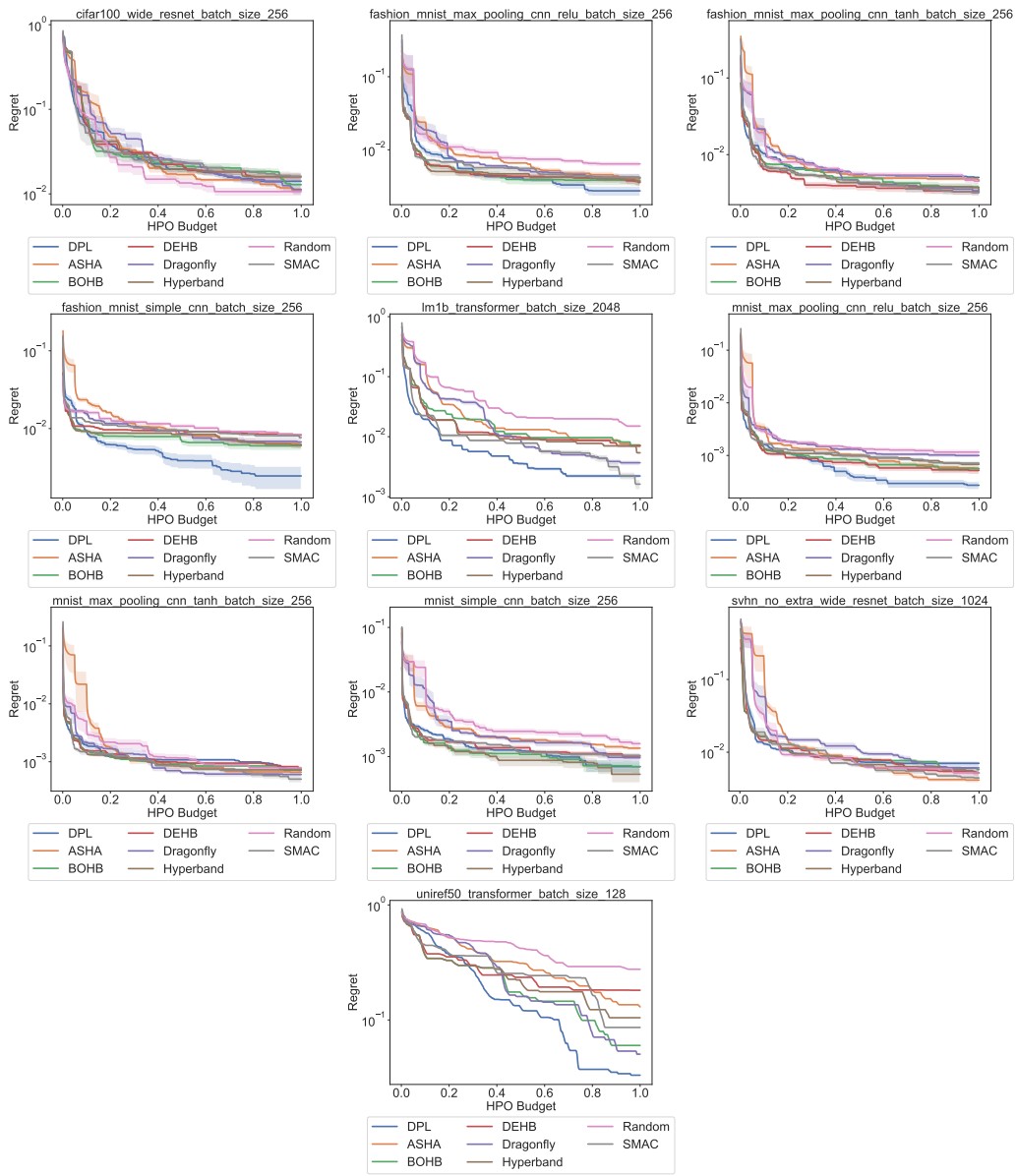

Figure 23: Performance comparison over the fraction of the total optimization iterations on a dataset level for PD1. We plot the mean value over 10 runs and the standard error.

