# OpenReview forum: "Scaling Laws for Hyperparameter Optimization"
_NeurIPS.cc/2023/Conference — NeurIPS 2023 poster_

### Official Review · Reviewer_ALT2 · 2023-07-01

**Soundness:** 3 good
**Presentation:** 3 good
**Contribution:** 2 fair
**Rating:** 5
**Confidence:** 1

**Summary:**

The authors present a hyperparameter tuning scheme based on optimizing surrogate powerlaws. They claim substantial improvements over baselines on hyperparameter tuning benchmarks.

**Strengths:**

The writing is generally clear, and the method is well described. The authors report substantial improvements over baselines on what appear to be standard benchmarks. Real improvement here would be significant for the field.

I should note here that I am unfamiliar with these HPO baselines and benchmarks. While the reported results seem potentially interesting, I place low confidence in my own assessment here. Input from other reviewers with more familiarity with the methods of assessment and the difficulties one encounters when using HPO in practice will be important here.

**Weaknesses:**

It is very hard to assess Hypothesis 1 from Figure 1. This figure seems like it ought to be a final summary figure after some more illustrative figures showing, for example, parameter vs. performance, with five proposed fits overlaid. More generally, the lack of figures explicitly showing powerlaws in this paper about powerlaws is quite odd and makes it hard to gauge the claims made. For example, I'm quite skeptical that *all* metrics one might want to track follow nice powerlaws as the proposed fitting function assumes -- which seems like it'd largely invalidate the proposed HP tuner -- but I could be convinced by lots and lots of plots showing that metric after metric is indeed forecasted nicely by a powerlaw. This is what I'd expect to see here.

For a paper using powerlaw scaling, there's a surprising lack of discussion of [Kaplan et al. (2020)](https://arxiv.org/abs/2001.08361) or the Chinchilla scaling laws, which are foundational results dealing with powerlaws + large models used in practice.

**Questions:**

On "Unfortunately, HPO is not yet feasible for Deep Learning (DL) methods" -- what does this mean? Hyperparameters are optimized all the time.

---

> ### Author Rebuttal · Authors · 2023-08-08
>
> We would like to thank the reviewer for the thoughtful review. We provide the following clarifications on the questions raised by the reviewer:
>
> - **Regarding “It is very hard to assess Hypothesis 1 from Figure 1. This figure seems like it ought to be a final summary figure after some more illustrative figures showing, for example, parameter vs. performance, with five proposed fits overlaid. More generally, the lack of figures explicitly showing powerlaws in this paper about powerlaws is quite odd and makes it hard to gauge the claims made.”:**
>
>     We would like to initially clarify to the reviewer that our power law surrogate is a function of the number of epochs/iterations of optimization (as explained in the experimental protocol in Section 5) and not on the number of parameters.
>
>     Hypothesis 1 Figure 1 is directly addressing the reviewer's question on “How well do power laws model learning curves”. Our experiment is based on the rationale that “method A models learning curves better than method B if method A is able to forecast future unobserved values of the learning curve (given a partially observed curve) more accurately than method B”. We opted for Hypothesis 1 because we believe that reporting the aggregated forecasting accuracy across thousands of learning curves is a more principled assessment than visually interpreting a few learning curves.  Regarding the metric of evaluating the forecasting performance, we present both results in mean absolute relative error in Appendix J, Figure 18, as well as the rank correlation in Figure 1.
>
>
>     Nevertheless, we understand the reviewer’s concern that simpler visualizations will help readers understand the method more quickly. As a result, we provide additional figures (extra page, in the global response) that visually show how DPL fits learning curves from diverse datasets included in our experiments. DPL is given partial observations from **only the learning curve that is investigated** and infers the rest of the learning curve based on the observed points.
>
>     We will incorporate these visualizations into the camera-ready version.
>
>
> - **Regarding “For example, I'm quite skeptical that all metrics one might want to track follow nice powerlaws as the proposed fitting function assumes -- which seems like it'd largely invalidate the proposed HP tuner”:**
>
>     We would refer the reviewer to [1], additionally [2] (that the reviewer suggests) which show that well-behaved learning curves “generally” follow a power law pattern.
>
>     However, the reviewer is right in stating that not “all” learning curves follow a power law assumption. We fully agree with that statement, and also **stress this point in the paper in lines 247-249 “The findings indicate that even though not all learning curves are power laws, most of them are, therefore a power law surrogate is efficient in forecasting the final performance of a partially-observed learning curve.”**
>
>
>     At the end of the day, the quality of the HPO results is the ultimate metric of success, not the quality of learning curve modeling. HPO is a different process than learning curve forecasting, because it focuses on exploring and exploiting regions of performant hyperparameter configurations, in order to recommend the “best configuration to evaluate next”. Through ample experiments, we show that Bayesian optimization equipped with our novel power law surrogate and acquisition achieves better HPO results.
>
>     It is worth emphasizing that learning curves that deviate significantly from the power law assumption are usually diverging configurations (e.g. loss and error rate increasing suddenly, as in the case of very high learning rates). Modeling such learning curves suboptimally appears to not hurt the quality of HPO, because divergent learning curves usually represent bad hyperparameter configurations with high error rates. Since in HPO with Bayesian optimization the acquisition recommends only the best-estimated configuration (the one with the lowest estimated error rate), it is not essential if we optimally or suboptimally estimate the learning curve of the configurations with a high error rate, because the bad configurations are not recommended for evaluation by the HPO algorithm.
>
>
>     **For transparency, we would point the reviewer to Lines 239-245 where we provide a few ways on how to further tackle learning curves that have a divergent behavior. As pointed out by the aforementioned lines, the analysis is extended in Appendix C in more detail.**
>
> - **Regarding: For a paper using powerlaw scaling, there's a surprising lack of discussion of Kaplan et al. (2020) or the Chinchilla scaling laws, which are foundational results dealing with powerlaws + large models used in practice.**
>
>     We agree with the reviewer. We will add the suggested papers to the list of already-cited prior works on scaling laws in Section 3 for the camera-ready version.
>
> - **Regarding: On "Unfortunately, HPO is not yet feasible for Deep Learning (DL) methods" -- what does this mean? Hyperparameters are optimized all the time.**
>
>     By HPO we refer to principled and automated searching techniques for tuning hyperparameters, such as Bayesian optimization. rather than manual and ad-hoc parameter tuning. For modern Deep Learning, most researchers and practitioners follow suboptimal trial-and-error searching of hyperparameters based on a local search around an initial guess of hyperparameter values.
>
> If the reviewer is pleased with the clarifications and proposed changes we would appreciate a reflection of the discussion to the score. In case there are more questions, we are happy to answer them.
>
> [1] Mohr, F., & van Rijn, J. (2022). Learning Curves for Decision Making in Supervised Machine Learning--A Survey. arXiv preprint arXiv:2201.12150.
>
> [2] Kaplan et al. Scaling laws for neural language models (2020)

---

### Official Review · Reviewer_wp1o · 2023-07-02

**Soundness:** 3 good
**Presentation:** 4 excellent
**Contribution:** 3 good
**Rating:** 7
**Confidence:** 3

**Summary:**

AFTER REBUTTAL: I acknowledge reading the rebuttal. After the reviewer discussion, I will keep my score the same.

----

Gray-box hyperparameter optimization involves optimizing neural network hyperparameters by evaluating performance at low budgets and terminating configurations if they seem unpromising. This paper proposes a gray-box scheme using Bayesian optimization over neural scaling laws to estimate future model performance. The key insight is to fit an ensemble of neural networks that can estimate the power law parameters and then apply BO to estimate new hyperparameter settings. Numerical results demonstrate the power law’s ability to forecast as well as state-of-the-art relative regret in HPO.

**Strengths:**

Overall the paper is well-written, clearly motivated, suggests an intuitive strategy, and rigorously experimented.

**Weaknesses:**

-	The function $f(\lambda)$ gets overloaded with $f(\lambda, b)$ or as $f(b)$ in various parts.
-	The definition of the cost function is not present. It would be nice to have a slower explanation of the experiment setup including the cost function and budget, since the plots tend to just look at normalized budget metrics.
-	The ideas of learning a meta model of power laws, of using Bayesian optimization over power laws, and using an ensemble of power laws have all been explored in recent works. Although these works do not focus on hyperparameter optimization, it would be nice to differentiate the current work from the previous in terms of methods.
 - Jain, Achin, et al. "A meta-learning approach to predicting performance and data requirements." Proceedings of the IEEE/CVF Conference on Computer Vision and Pattern Recognition. 2023.
 - Tejero, Javier Gamazo, et al. "Full or Weak annotations? An adaptive strategy for budget-constrained annotation campaigns." Proceedings of the IEEE/CVF Conference on Computer Vision and Pattern Recognition. 2023.
 - Mahmood, Rafid, et al. "Optimizing data collection for machine learning." Advances in Neural Information Processing Systems 35 (2022): 29915-29928.

**Questions:**

Please see the weaknesses

**Limitations:**

Limitations are not meaningfully discussed.

---

> ### Author Rebuttal · Authors · 2023-08-08
>
> We would like to thank the reviewer for the thoughtful review. We provide the following clarifications on the questions raised by the reviewer:
>
> - **Regarding “The function $f\left(\lambda\right)$ gets overloaded with $f\left(\lambda, b\right)$ or as $f\left(b\right)$ in various parts.”:**
>
>     We thank the reviewer for spotting the inconsistency in our formalism, to improve clarity, we will rephrase $f^{best}\left(b\right)$ and $f^{oracle}\left(b\right)$ as $f\left(\lambda^{best}, b\right)$ and $f\left(\lambda^{oracle}, b\right)$. This affects Equation 7 (bottom) and Equation 9 (as well as the text that refers to both terms in Line 126 and Lines 146-151). We will incorporate the aforementioned changes for the camera-ready version of our work.
>
>
>     Regarding the preliminaries of Section 3, there is no notion of $b$ for the first 2 paragraphs as multi-fidelity has not been introduced yet. Additionally, in Section 6, when $f\left(b\right)$ is used, it refers to models that do not take the hyperparameter configuration as an input.
> - **Regarding "The definition of the cost function is not present. It would be nice to have a slower explanation of the experiment setup including the cost function and budget, since the plots tend to just look at normalized budget metrics.":**
>
>     We would like to point the reviewer to the **experimental protocol, Section 5, lines 137-144** where we describe the cost function and budget. **“One unit step of the HPO budget signifies training one hyperparameter configuration for one more step (1 epoch in LCBench, or 200 iterations in TaskSet)”**. We then describe in detail what step represents for all benchmarks and the benchmark-specific default evaluation metric. We summarize the information below:
>
>     |Benchmark | Learning curve step |Evaluation metric|
>     |:---------------|:---------------------------|:----------------------|
>     |LCBench    | 1 epoch                    | Bal. Accuracy      |
>     |TaskSet      | 200 batch iterations  | Loss                   |
>     |PD1            | 1 epoch                     | Accuracy           |
>
> - **Regarding "The ideas of learning a meta-model of power laws, of using Bayesian optimization over power laws, and using an ensemble of power laws have all been explored in recent works. Although these works do not focus on hyperparameter optimization, it would be nice to differentiate the current work from the previous in terms of methods.":**
>
>     We thank the reviewer for suggesting the parallel works at CVPR 2023, and the NeurIPS 2022 paper, which although do not address HPO, still elaborate on scaling laws for performance predictions. We will integrate the suggested works into our work for the camera-ready version. We agree with the comments on novelty, **except for the point “... using Bayesian optimization over power laws … has been explored in recent works”**. To the best of our awareness, we are the first to explore Bayesian optimization for HPO with power laws.
>
> We believe to have adequately addressed the questions from the reviewer. In case there are more questions, we are happy to answer them

---

### Official Review · Reviewer_rbxf · 2023-07-03

**Soundness:** 4 excellent
**Presentation:** 4 excellent
**Contribution:** 3 good
**Rating:** 7
**Confidence:** 3

**Summary:**

The paper proposes a novel surrogate for multi-fidelity HPO that uses an ensemble of power-law models to estimate the future validation loss of hyper-parameter configurations at intermediate stages of training. The principal novelty of the work lies in exploiting the observation (made in other work) that learning curves tend to follow power laws.
Thorough experiments are presented across diverse datasets and tasks, comparing against publicly available strong HPO baselines; SOTA performance is demonstrated. Analysis demonstrates an improvement in learning-curve forecasting for the surrogate over non-power law techniques, as well as an increase in efficiency for DPL over HPO baselines.


**Strengths:**

The paper presents a novel contribution to the HPO methodology literature. The claims of the paper are well supported by experiment, and generally well analyzed. The specification of the separate hypotheses of the work in the analysis gives a pleasantly clear structure to the paper. The results demonstrably advance the SOTA in HPO; this work seems very likely to be used and built upon in future HPO work.

**Weaknesses:**

The final analysis section, evaluating the use of DPL for LLMs, is less convincing than the previous sections. Significant work is done by line 319 “We follow the common practice of conducting HPO with small transformers and then deploying the discovered optimal configuration on the full-scale transformers”. Citations of prior work which discuss this convention or apply the ‘common practice’ should be included (perhaps the paper on Tensor Programs V (Yang et al.2022)). Following the convention of that paper, it would be valuable to include the total computational cost of tuning the small models as a proportion of the computational cost of training the large model once (with the small-model-identified parameters). This would give a much better sense of the practicality of applying such a technique in a LLM setting, where the computational burden of training is paramount.

Nits:
In Section 5.1, the architectures of LCBench and PD1 are mentioned, but for TaskSet no mention of architecture is made.
In the LLM section, total parameter count of the various models should be reported.

**Questions:**

The results on PD1 seem weaker than those for LCBench and TaskSet in Figures 2,3,4, and PD1 is left out of Figure 5 and pushed to the appendix. Do you have any analysis or commentary on the relative performance of the method as a function of the architectures or tasks involved in each benchmark? From the description in Sec 5.1, it seems PD1 is a quite diverse benchmark. While the analysis presented quite reasonably focuses on the aggregated performance of each benchmark, was any analysis performed as to the relative performance on specific architectures, tasks, or hyper-parameter spaces?

**Limitations:**

The practical limitations of the applicability of this work towards LLMs may be under-explored. While the extension of the method towards giant models is not necessary for the impact of this generally strong contribution, it remains unclear if the efficiency gains over existing HPO methods are sufficient to overcome the general concerns of computational inefficiency which have typically precluded Bayesian HPO methods from general use in larger models.

---

> ### Author Rebuttal · Authors · 2023-08-08
>
> We would like to thank the reviewer for the thoughtful review. We provide the following clarifications on the questions raised by the reviewer:
>
> - **Regarding “Significant work is done by line 319. We follow the common practice of conducting HPO with small transformers and then deploying the discovered optimal configuration on the full-scale transformers. Citations of prior work which discuss this convention or apply the ‘common practice’ should be included (perhaps the paper on Tensor Programs V (Yang et al.2022))“:**
>
>
>     We agree with the reviewer. To improve clarity, we will refer again to the Tensor Programs V work from Line 74 (in the related work).
>
> - **Regarding "Following the convention of that paper, it would be valuable to include the total computational cost of tuning the small models as a proportion of the computational cost of training the large model once (with the small-model-identified parameters)."":**
>
>     We agree with the reviewer. In our LLM experiment (Section 6, Hypothesis 4), it takes 3.66 hours to find the Oracle configuration for the largest model via HPO for the smallest model. In turn, it takes 21.52 amount of hours to train the largest model only once. As such, the proportion is 0.17. We will update our camera-ready version as suggested by the reviewer.
>
> - **Regarding: "In Section 5.1, the architectures of LCBench and PD1 are mentioned, but for TaskSet no architecture is mentioned.":**
>
>     We thank the reviewer for pointing out the missing information. The architectures for TaskSet are a Variational RNN, Identity RNN, GRU RNN, and LSTM RNN. We will make sure to update Section 5.1 for the camera-ready version with the details of the architectures included in TaskSet.
>
> - **Regarding: "In the LLM section, the total parameter count of the various models should be reported."**
>
>     We agree with the reviewer. The parameter counts for the various models are as follows:
>
>     | Embedding Size  | Total Parameters |
>     | :---------------------- | :---------------------: |
>     | 6                           | 0.3 M                   |
>     | 12                         | 0.6 M                   |
>     | 24                         | 1.2 M                   |
>     | 48                         | 2.6 M                   |
>     | 96                         | 5.5 M                   |
>     | 192                       | 12.4 M                 |
>     | 384                       | 30.1 M                 |
>
>     We will update the camera-ready version to include the parameter counts of the models.
> - **Regarding "The results on PD1 seem weaker than those for LCBench and TaskSet in Figures 2,3,4, and PD1 is left out of Figure 5 and pushed to the appendix. Do you have any analysis or commentary on the relative performance of the method as a function of the architectures or tasks involved in each benchmark? From the description in Sec 5.1, it seems PD1 is a quite diverse benchmark. While the analysis presented quite reasonably focuses on the aggregated performance of each benchmark, was any analysis performed as to the relative performance on specific architectures, tasks, or hyper-parameter spaces?"**
>
>     We thank the reviewer for the interesting question. We would like to point the reviewer to Section 6, Hypothesis 2 (Line 274) where we provide our comments on the lack of statistical significance for PD1.
>
>
>
>     As a summary, we did investigate the specific tasks/search spaces where DPL does not perform as well as the other baselines, as the reviewer suggests. We notice that the tasks have a skewed distribution of hyperparameter configuration performances, where, a majority of the configurations achieve top performance.
>
>     The detailed analysis can be found in Appendix F.
>
>
> We believe to have adequately addressed the questions from the reviewer. In case there are more questions, we are happy to answer them.

---

> > ### Comment · Reviewer_rbxf · 2023-08-14
> >
> > Thanks to the authors for the thorough response. All of the questions I had have been adequately addressed. On reading the other reviews and responses, I maintain my vote for acceptance.

---

### Official Review · Reviewer_sM9u · 2023-07-05

**Soundness:** 3 good
**Presentation:** 2 fair
**Contribution:** 3 good
**Rating:** 6
**Confidence:** 3

**Summary:**

This paper proposes to combine power law patterns in learning curves, which have been recently popularized by scaling laws, to improve efficiency and performance of Bayesian optimization (BO) based hyperparameter optimization (HPO). Authors provide detailed discussions on how to model the surrogate function and accuracy of their power law based surrogate function in predicting final performance. Tested on three HPO benchmarks, authors empirically demonstrate that DPL achieves better performance given a limited budget compared to other BO-based HPO baseline algorithms.

**Strengths:**

1. I believe the strategy of incorporating power law patterns from scaling laws into the surrogate function of Bayesian optimization is neat and smart in that it enables an improved prediction of learning curves.
2. In addition, authors have empirically justified the soundness of their method through diverse analyses and experiments.
3. Given its simplicity, I expect this method would generalize across different tasks not explored in this paper.
4. Authors described their experiment settings in detail, which was very helpful in evaluating the significance of their method.

**Weaknesses:**

1. I believe the clarify of the paper could be potentially improved. For example, abstract of this paper is too simple, and I wasn't able to grasp the general direction or the concept of the paper by reading them. Also, I believe moving the algorithm box from Appendix to the main text would help readers to better understand their method. I currently see quite a lot of empty spaces in the paper, so expect, with minor formatting efforts, it would be easy to move the algorithm box to the main text.

**Questions:**

1. I am curious about the applicability of DPL to different ML techniques such as random forest or XGBoost, which is arguably the most popular option for tabular data. I am not an expert in this domain, but I doubt the concept of learning curves also plays an important role in those types of models. If not, I believe stating it clear and focusing writing on the area where their method shines most would actually improve the clarify of the paper.
2. I want to know author's opinion on the applicability of their method on training recent large models. The training cost of recent large models can be astronomical, so multiple training runs assumed by Bayesian optimization may not be realistic despite DPL's improved efficiency compared to the baseline. Even though authors included some transformer experiments, I still think the size of their transformers are quite smaller than recent models. This question is not specifically about DPL, but more about BO-based HPO. Any insights would be much appreciated.

**Limitations:**

Authors haven't clearly discussed the limitations of their work. Having one or two sentences on the limitations would be helpful for people who are interested in this direction.

---

> ### Author Rebuttal · Authors · 2023-08-08
>
> We would like to thank the reviewer for the thoughtful review. We provide the following clarifications on the questions raised by the reviewer.
> - **Regarding “the abstract of this paper is too simple, and I wasn't able to grasp the general direction or the concept of the paper by reading them.”:**
>
>     We will improve the abstract in two ways for the camera-ready version by better describing: i) the problem (describing the multi-fidelity HPO approach with learning curves) we are addressing and ii) the proposed method (introducing a probabilistic scaling law performance predictor with a dynamic acquisition function for Bayesian optimization).
>
>     If the reviewer has other suggestions we would be happy to incorporate them.
> - **Regarding “I believe moving the algorithm box from Appendix to the main text would help readers to better understand the method”:**
>
>     We agree with the reviewer’s suggestion and will move the algorithm to the main paper for the camera-ready version.
>
> - **Regarding the “applicability of DPL to different ML techniques such as random forest or XGBoost”:**
>
>     The reviewer is correct. It would be possible to apply DPL for ML techniques such as random forest or XGBoost, by considering the number of ensemble models as a budget/fidelity. This is a very interesting future work that we will add to a new section “Limitations and Future Work”.
>
> - **Regarding the "the applicability of the DPL method in training recent large models":**
>
>     Searching directly on a model with billions of parameters might be indeed challenging. However, in practice, we can tune the hyperparameters on a smaller version and transfer it to the larger version.
>
>     We would like to refer the reviewer to [1] who motivated a possible transfer theoretically and demonstrated empirically that models with **a few million parameters transfer to LLMs with billions of parameters**. Other AutoML works employ the same idea. Most cell-based NAS methods (for example the following early work [2]) are training small networks and later scale them by increasing width and depth. Similarly, work on transformer search makes use of the same technique [3].
>
> [1] Yang, Ge, et al. "Tuning large neural networks via zero-shot hyperparameter transfer." Advances in Neural Information Processing Systems 34 (2021): 17084-17097.
>
> [2] Pham, Hieu, et al. "Efficient Neural Architecture Search via Parameter Sharing." ICML 2018: 4092-4101
>
> [3] So, David, et al. "The Evolved Transformer." ICML 2019: 5877-5886
>
> If the reviewer is pleased with the clarifications and proposed changes we would appreciate a reflection of the discussion to the score. In case there are more questions, we are happy to answer them.

---

> > ### Comment · Reviewer_sM9u · 2023-08-10
> > **Thanks for the rebuttal**
> >
> > Thanks for the rebuttal with clarifications. I maintain my score, voting for the acceptance. Good luck!

---

### Official Review · Reviewer_7buU · 2023-07-06

**Soundness:** 4 excellent
**Presentation:** 3 good
**Contribution:** 3 good
**Rating:** 7
**Confidence:** 4

**Summary:**

The paper proposes deep power laws ensembles for hyperparameter optimization. More precisely, it is used as a surrogate for Bayesian Optimization (BO) to estimate the performance of a hyperparameter configuration leveraging ensembles of deep power law functions. Furthermore, it is combined with multi-fidelity optimization to estimate the performance for an upcoming budget, enabling incremental training that can be paused for individual configurations based on the performance estimate of the surrogate.

**Strengths:**

### Originality and Quality
The presented approach is interesting, as it combines promising directions of HPO: Multi-Fidelity and Bayesian Optimization based on power laws. It is original in the way that it is the first work (to the best of my knowledge) that leverages Deep Ensembles for power laws as surrogates for BO.Moreover, a simple but original multi-fidelity strategy is presented to dynamically adjust the budget a configuration is evaluated on, allowing incremental training of configurations by pausing and continuing the process dependent on the performance estimate of the surrogate. Overall, none of the parts of the presented approach is innovative on its own, but the combination thereof is.

### Clarity
The paper is well structured and written in an understandable manner. The different components and their interaction could be clearly separated and emphasized more.

### Significance
The presented method is evaluated on 3 state-of-the-art benchmarks, as well as compared to 7 state-of-the-art HPO tools. The evaluation of 4 well selected hypotheses lead to the conclusion that assumptions are accurate, the approach is working correctly, and the presented method outperforms competitors in most cases.


**Weaknesses:**

### Originality
The originality of this paper is partly hard to define, as related work or citations are missing at key aspects of this paper. Especially regarding the usage of power laws in the context of HPO, existing work is not investigated / cited [1, 2]. Furthermore, motivation, explanation, and limitation of multi-fidelity and power laws are mixed up (see e.g. l. 57-59). Moreover, the presented multi-fidelity strategy is not put into context of existing strategies. Furthermore, the authors mix up gray-box HPO and multi-fidelity HPO (e.g. l. 95: “Gray-box (multi-fidelity)”). Multi-fidelity can be classified as gray-box, but not the other way around.  In addition, the assumption that every learning curve can be described by a power law function should be verified by a reference or two. Overall, the work could be better embedded into existing work on learning curves, see [2].

### Clarity and Quality
The clarity and quality of the paper can be improved. On one hand, the paper should be self contained, which mainly refers to a missing formal introduction and short background of power law functions. Related work and approach description are partially mixed or not suitably placed, e.g. exploiting a power law assumption with the presented method would have been expected to be explained within the description of the method and not as part of related work of multi-fidelity HPO. On the other hand, the mathematical formulation can be improved. Details below:
1. l. 52-53: “[...] the budget is multiplied by the fraction of discarded hyperparameter configurations and the process continues until the maximum budget is reached.” This would reduce the budget with every step, but it should be increased to reach the maximum budget. Suggestion: “[...] budget is divided by the fraction [...]”.
2. l. 57-59: “However, the only assumption these methods make about the learning curve is that it will improve over time. In contrast, we fit surrogates that exploit a power law assumption on the curves.” There is already work leveraging power-laws for performance prediction of learning curves [1, 2].
3. l. 79f / Eq. 1: Too much space between $\theta^*$ and the next bracket
4. l. 82: The basic definition of $H$ should contain $N$, as it is defined later in Eq. 2 (l. 87f), otherwise multiple usages of $H$ are ambiguous.
5. l. 83: $\mathcal{A}$ is defined, and directly afterwards the normal $A$ is used. Furthermore, the set definition of all possible $\mathcal{A}$ is missing which should be used for $\arg\min$ (in l. 87f)
6. l. 87f / Eq. 2: “$\lambda_i = \mathcal{A}$” is an unusual notation and more common in programming than math, “$\lambda_i =$” could be removed here to avoid confusion.
7. l. 87f / Eq. 2: $\Omega$ should be $\geq$ (greater equals) the sum of the cost, as the budget should not be exceeded, but can be matched.
8. l. 91: $p$ is not defined, as well as the set definition of all possible $\phi$.
9. l. 138: What is “the loss”?
10 l. 201: SMAC is not necessarily an extension of HB, but only the mf part of SMAC uses HB. This should be properly described.
Lastly, there is a grammar issue in l. 55: “elaborated” → “elaborated on”, and Figure 5 is misplaced as it belongs to the earlier section.

[1] Buratti, B., & Upfal, E. (2019). Ordalia: Deep Learning Hyperparameter Search via Generalization Error Bounds Extrapolation. In 2019 IEEE International Conference on Big Data (Big Data) (pp. 180-187). IEEE.

[2] Mohr, F., & van Rijn, J. (2022). Learning Curves for Decision Making in Supervised Machine Learning--A Survey. arXiv preprint arXiv:2201.12150.


**Questions:**

For easier reference during the rebuttal, the questions are enumerated below:
1. Do you agree or disagree with any of the remarks from section Weaknesses?
2. Do you agree or disagree with the suggested corrections of the mathematical formulation listed in the Weaknesses section?
3. l. 171-172: “HPO budget is defined as the maximum number of steps needed to fully evaluate 20 hyperparameter configurations.” How about the overall CPU hours needed to execute the experiments with different approaches? Do they differ? Are there any additional ecological insights, e.g. from a Green AutoML perspective [3], like one might save wall clock time compared to others? Does this relate to performance improvement?
4. l. 205-206: The description of the used hardware is incomplete as memory is missing.
5. I am wondering why there is no bold conclusion for Hypothesis 4?

Final remark regarding the overall rating: I am strongly convinced that the weaknesses can be easily addressed during the rebuttal phase. If the authors do so, I am more than happy to increase my score to an accept as I think that the paper is really good besides the weaknesses above.

[1] Buratti, B., & Upfal, E. (2019). Ordalia: Deep Learning Hyperparameter Search via Generalization Error Bounds Extrapolation. In 2019 IEEE International Conference on Big Data (Big Data) (pp. 180-187). IEEE.

[2] Mohr, F., & van Rijn, J. (2022). Learning Curves for Decision Making in Supervised Machine Learning--A Survey. arXiv preprint arXiv:2201.12150.

[3] Tornede, T., Tornede, A., Hanselle, J., Mohr, F., Wever, M., & Hüllermeier, E. (2023). Towards Green Automated Machine Learning: Status Quo and Future Directions. Journal of Artificial Intelligence Research, 77, 427-457.


**Limitations:**

Limitations are not explicitly given in the paper if I have not overseen something.

---

> ### Author Rebuttal · Authors · 2023-08-08
>
> We would like to thank the reviewer for the thoughtful and detailed review of our work. We provide the following clarifications on the questions raised by the reviewer.
>
> - **Regarding the remarks from the section Weaknesses**:
>     - **In terms of related work:**
>
>         We will cite both [1] and [2] in our related work.  Additionally, in the case of [1], we will highlight the difference in terms of proposing a
>         conditioned probabilistic power law surrogate that needs no observations per LC in order to estimate the performance. The ability to
>         estimate the performance of unobserved configurations through probabilistic surrogates is essential for Bayesian optimization,
>         therefore, making our method the first to propose power law surrogates for uncertainty-driven HPO.
>     - **Regarding “motivation, explanation, and limitation of multi-fidelity and power laws are mixed up (see e.g. l. 57-59)”:**
>
>         We understand the reviewer’s concern, however, we believe the related work is structured in a consistent manner. At the end of every
>         related work paragraph, we have a sentence that delineates the novelty of our paper from the prior work of that paragraph. In terms of
>         multi-fidelity HPO, our novelty is in proposing a novel surrogate with the power law assumptions. This is the reason why the sentence
>         refers to both multi-fidelity HPO (problem definition) and power laws (novelty).
>     - **Regarding “the presented multi-fidelity strategy is not put into context of existing strategies” and “Overall, the work could be better embedded into existing work on learning curves, see [2].”:**
>          Our work is positioned as in the context of well-established multi-fidelity HPO (problem definition) and Bayesian optimization (BO, strategy for solving the problem). Given our novel power-law surrogate and acquisition, which are very standard BO components, we assess the method to be positioned within a well-established BO strategy. Additionally, the surrogate function with power laws is connected to existing work on **learning curves (Line 60)**. Furthermore, we cite multiple papers in the learning curves prediction paragraph, the majority of which are cited by [2]. However, we agree with the reviewer in further extending/strengthening the related work with the aforementioned works [1]-[2].
>     - **Regarding “mix up gray-box HPO and multi-fidelity HPO (e.g. l. 95: “Gray-box (multi-fidelity)”)”:**
>
>         We agree with the reviewer. We will make it clear in our work that we address multi-fidelity HPO, and specify that it is a sub-problem of gray-box HPO as the reviewer suggests.
>
> - **Regarding the "suggested corrections of the mathematical formulation":**
>
>     We agree with the suggested changes on the formalism from the reviewer and we will update the camera-ready accordingly.
>
> - **Regarding the "time performance and ecological insights":**
>
>     The reviewer is correct in his/her understanding that the budget for the wallclock experiments is the runtime equivalent of fully evaluating 20 randomly selected configurations. This budget limit is set as a practical compromise considering our available computational resources. All methods, therefore, need to search for an incumbent (best per method) configuration within this budget. For instance, if 20 randomly selected hp configurations are evaluated for a total of say 100 hours, then we set the HPO budget for all methods to 100 hours. Within this budget, methods are free to partially or fully evaluate as many configurations as they decide within their search mechanisms. The best model's performance discovered by each method within the budget is used to compute the comparative metrics; namely regret and ranks.
>
>     Regarding the wallclock HPO time, we point the reviewer to **Hypothesis 2 L283-289 (Figure 4)** where we fairly compare all methods over time [surrogate fit (if it is a model-based method) + hyperparameter evaluation]. **The setup for the experiment is described in Section 5 L159-166**. Summarized, the time of every algorithm is normalized by the random search time and the total time shown in Figure 4 is the time it took random search to perform 20 HPO trials.
>
>     Regarding ecological insights, our method uses fewer computational resources to achieve the same/better performance compared to other methods.  In the case of LCBench, our method matches the final performance of random search in approximately 10% of the total time and the final performance of the closest competitor (BOHB) in 20% of the total time. While for PD1, our method matches the final performance of random search in 30% of the total time and matches the performance of DragonFly the closest competitor in 80% of the total time.
>
> - **Regarding "the memory description of the used hardware":**
>
>     We would like to thank the reviewer for pointing out the missing information. The total memory of every node is 120GB, and every experiment is limited to 2 cores which offer 12GB. We are going to modify Section 5 accordingly for the camera-ready version of our paper since the rebuttal phase does not allow modifications to the submitted manuscript.
>
> - **Regarding "the missing bold conclusion for Hypothesis 4":**
>
>     We agree with the reviewer, and we will add a bold conclusion for Hyp. 4 along the lines of "The results validate Hypothesis 4 and confirm that DPL is an efficient HPO technique for tuning the hyperparameters of large language models when the HPO is conducted using smaller transformer model sizes."
>
>
> If the reviewer is satisfied with the clarifications and proposed changes we would appreciate a reflection of the discussion to the score. In case there are more questions, we are happy to answer them.

---

> > ### Comment · Reviewer_7buU · 2023-08-14
> > **Score Increased**
> >
> > Thank you very much for the detailed response! Since the authors answered all of my questions and will adjust for the remaining points, I am more than happy to increase my score.

---

### Author Rebuttal · Authors · 2023-08-08

We would like to thank all the reviewers for the thorough reviews and for helping us improve the quality of our work. Below, we summarize the main clarifications:

- **Time performance and ecological insights (Reviewer 7buU):** Hypothesis 2 of Section 6 provides a comparison between all methods for the total time it took random search (a model-free method) to evaluate 20 hyperparameter configurations, where our method achieves better or same results compared to our competitors with less computational resources.
- **Weaker results in the PD1 benchmark (Reviewer rbxf):** Hypothesis 2 of Section 6 provides insights on the lack of statistical significance in the PD1 benchmark. A detailed analysis is provided in Appendix F.
- **Missing definition of the cost function and budget (Reviewer wp1o):** The experimental protocol of Section 5.3 provides information in detail on the cost function of experiments (learning curve step or time). It additionally provides information on the default evaluation metric in every benchmark.
- **Explicit mention of the limitations:** We will add a new section to our work labeled “Limitations and Future Work”.

We believe to have answered all of the questions raised by the reviewers and we welcome any new questions during the discussion period.

---

### Decision · Program_Chairs · 2023-09-21

**Decision:**

Accept (poster)

**Comment:**

The reviewers and meta reviewer all carefully checked and discussed the rebuttal. They thank the authors for their response and their efforts during the rebuttal phase.

The reviewers and meta reviewer all acknowledge the original and sound contribution of exploiting a power-laws parametrization with deep ensembles for HPO. Similarly, the reviewers and meta reviewer all call out the overall good quality of the manuscript (writing, structure).

The rebuttal has considerably strengthened some aspects of the work (e.g., richer discussion of related work). However, there are still some remaining important concerns. As a result, the reviewers and the meta reviewer are weakly inclined to accept the paper.

In particular, the authors are urged to carefully update their final manuscript with the following points:

* Integrate in the final manuscript all the changes and discussions mentioned during the rebuttal phase
* Curically, the paper relies on deep ensembles for the surrogate model. This choice should be further motivated (e.g., it makes it possible to easily capture the uncertainty of the power-laws parametrization). Moreover, this choice brings up many important design decisions. In the current form, there are plenty of essential descriptions of those design decisions that are not present in the core manuscript (but rather hidden in the appendix or the code). _All those descriptions should instead be surfaced in the core manuscript_ (to that end, space can be gained by shortening/moving sections such as “preliminaries”). This includes (but it is not limited to)
    * Discussion about the architecture of ensemble members
    * Discussion about the retraining procedure when adding new observations (e.g., the code mentions some example-weighting and early-stopping schemes)
    * Discussion of the robustness of the approach at the beginning when there are a handful of observations in H (e.g., how many random guesses, how to train the deep ensembles with so little data, ….)
    * Optimisation of the acquisition function (e.g., the appendix/code seem to indicate that this is not done in a gradient-based fashion—how? why?)
* Missing ablation studies:
    * The effect of the only strategy in Eq. (8)
    * There is some previous research about deep ensembles for HPO, e.g.,  [A, B] (and [C], though more in RL-type settings). If the current submission makes design choices (see thread above) not supported by previous research, they need to be validated by some experiments/ablation studies. For instance, effect of the architecture, effect of the design of the retraining strategy, effect of the number of initial random guesses, etc. (in terms of organization, the discussion could happen in the core manuscript, as explained in the previous bullet point, and the ablation studies in the appendix)

If the paper was submitted to a journal, it would be accepted conditioned on those key changes, the meta reviewer thus expects all those changes to be carefully implemented.

[A] Salam Khazi et al., 2023, Deep Ranking Ensembles for Hyperparameter Optimization

[B] Kim et al., 2022, Deep Learning for Bayesian Optimization of Scientific Problems with High-Dimensional Structure

[C] Moberg et al., 2019, Bayesian Linear Regression on Deep Representations